# Evaluating the effects of topography and land use change on hydrological signatures: a comparative study of two adjacent watersheds

Haifan Liu[1], Haochen Yan[1], Mingfu Guan[1*]

[1]Department of Civil Engineering, the University of Hong Kong, Hong Kong, China

*Correspondence to*: Dr. Mingfu Guan (mfguan@hku.hk)

**Abstract.** Watershed hydrological processes are significantly influenced by land use/land cover change (LULCC) and characteristics such as topography. In economically advanced regions, coordinating land use planning and water resource management is essential for mitigating flood risks and ensuring sustainable development. This study compares the effects of terrain slope and urbanization-driven LULCC on hydrological processes in two adjacent subtropical watersheds but with distinct terrain and land cover in Greater Bay Area (GBA) of China. We developed an Integrated Surface-Subsurface Hydrological Model (ISSHM) using the Simulator for Hydrologic Unstructured Domains (SHUD) and calibrated it with data from river and groundwater monitoring stations. The calibrated model simulated hydrological processes including surface runoff, subsurface flow, evapotranspiration (ET), and infiltration to quantify water movement (measured in meters) and assess the impacts of slope and LULCC. Results show that slope impacts hydrological processes differently based on watershed characteristics. In mountainous areas, there are consistent high correlations between slope and annual surface runoff, infiltration, and subsurface flow across all watersheds. However, at lower elevations, the hydrological responses of steeper watersheds correlate weakly with local slope. Urbanization, marked by increased impervious surfaces, significantly raises annual surface runoff and decreases infiltration and ET, especially in steeper watersheds. In flatter watersheds, the rise in surface runoff is proportionally less than the increase in impervious areas, indicating a buffering capacity against urbanization. However, this buffering capacity is diminishing with increasing annual rainfall intensity. Overall, ISSHM provides robust analysis of LULCC effects on watershed hydrology across scales, enabling predictive approaches to optimize urban management for sustainable development in growing cities.

## 1 Introduction

The effects of land use/land cover change (LULCC) and topographic variability on hydrological processes within a watershed are widely recognized as critical issues in hydrology (e.g., Bosch and Hewlett, 1982; O'Loughlin, 1986; Costa et al., 2003; Beven, 2011; Gwak and Kim, 2016; Larson et al., 2022; Sicaud et al., 2024). Urbanization has been demonstrated to

significantly impact hydrological processes such as surface runoff, evapotranspiration (ET), infiltration, and subsurface flow

by altering the conditions of the land surface (Olang and Fürst, 2011; Ayalew et al., 2015; Guan et al., 2015; Bai et al., 2020;

Yan et al., 2023; Liang and Guan, 2024). Furthermore, it is evident that topographic characteristics have a direct influence on

surface water flow paths and soil moisture, thereby affecting infiltration rates and groundwater recharge (Strahler, 1957; Hopp

and McDonnell, 2009; Mirus and Loague, 2013; Smith et al., 2018; Yang et al., 2019; Zhang et al., 2022a). However,

comprehending the diverse impacts of LULCC and topography on hydrological processes across disparate watersheds persists

as a significant challenge, due to the variability in watershed characteristics and the nonlinear nature of hydrological responses

(Niehoff et al., 2002; Brath et al., 2006; Thanapakpawin et al., 2007; Du et al., 2012; Pang et al., 2022; Yin et al., 2023; Guo

et al., 2023; Yan et al., 2024). In order to address these challenges, researchers employ various methodologies to dissect and

quantify these effects.

Statistical analysis techniques utilizing long-term monitoring data within a watershed are commonly used to examine the

effects of LULCC (Beven et al., 2008; Liu et al., 2017; Zhang et al., 2021; Zhang et al., 2022b; Kumar et al., 2022). However,

long-term changes in hydrological responses often reflect the combined impacts of climate change and LULCC, making it

complicated to isolate the impacts of LULCC (Beven, 2011). The paired catchments approach is another statistical method

commonly employed (Brown et al., 2005; Detty and McGuire, 2010; Yang et al., 2016; Van Loon et al., 2019), which compares

monitoring data from two watersheds with different land cover but similar physical characteristics (Li et al., 2009; Shao et al.,

2020). However, applying this approach in practice can be challenging due to the difficulty in identifying watersheds with

similar physical characteristics. Furthermore, recent studies have indicated that LULCC-induced hydrologic alterations exhibit

considerable spatial variability within watersheds, affecting upstream and downstream regions in disparate ways (Chu et al.,

2010; Garg et al., 2017). In this regard, statistical analysis methods that rely on gauging datasets often lack detailed spatial

resolution, employing methods that facilitate studies at finer spatial resolutions is essential for a comprehensive understanding

of these variations.

Similar challenges exist when investigating the effects of topography on watershed-scale hydrological processes due to

the diversity of geomorphic types and significant spatial variability within watersheds. One area where significant progress

has been made is the prediction of hydrologic connectivity through topographic indices to study rainfall-runoff responses in

watersheds (Jencso and McGlynn 2011). Topographic indices have become valuable tools for predicting soil moisture and

identifying saturated zones. Two successful examples are topographic wetness index (TWI; Beven and Kirkby, 1979; Sørensen

et al., 2006) and height above the nearest drainage (HAND; Nobre et al., 2011; Gao et al., 2019; Fan et al., 2019). However,

some studies reported TWI and groundwater levels exist distinct relations at different locations (Detty and McGuire, 2010;

Rinderer et al., 2014). Furthermore, the simulation results of HAND are highly depend on the pattern of observed saturated

zones and it perform better at gentler watersheds (Nobre et al., 2011; Gao et al., 2019). In addition, the predictive accuracy of these indices decreases under dynamic conditions, such as at the onset of rainfall events (Seibert et al., 2003; Jarecke et al., 2021).

Recent studies have shown that hydrological models based on the Richards equation not only simulate surface-subsurface water interactions on hillslopes but also accurately describe hydrological processes under varying temporal conditions (Camporese et al., 2019). The Integrated Surface-Subsurface Hydrological Model (ISSHM) is a type of Richards equation-based fully distributed hydrological model (Shen and Phanikumar, 2010; Maxwell et al., 2014; Fatichi et al., 2016). Despite being relatively new compared to other hydrological models, the ISSHM has demonstrated significant capabilities in addressing the whole system of processes at watershed scales (Niu et al., 2017; Yu et al., 2022; Zanetti et al, 2024). By dividing the land surface into grids, such models can represent the spatial variability of hydrological processes with high spatial accuracy. They can also be solved with higher temporal accuracy by applying differential solutions to the physical governing equations. Unlike monitoring data analysis methods, ISSHMs allow hydrologists to assess the impact of specific factors by implementing designed scenarios and evaluating them across a comprehensive range of spatial and temporal scales. In recent years, ISSHMs have been proven valuable for assessing LULCC and topographic impacts at the watershed scale. For instance, Im et al. (2009) used the MIKE SHE model to show that urbanization increased total runoff by 5.5% and overland flow by 24.8% in a watershed. Zhang et al. (2022a) explored how topography influences subsurface flow with the HydroGeoSphere, revealing that topography plays a significant role in controlling penetration depths and stagnant zones.

While some studies have investigated the effects of LULCC and topography using the ISSHM approach, they are primarily based on the single watershed (Chu et al., 2010; Im et al., 2009; Thanapakpawin et al., 2007), hindering comparative analyses. Herein, we showcase the behavior of paired watersheds with heterogeneous patterns of both terrains and land cover, but are geographically adjacent to be compared under the same subtropical climate regime. We simulate the hydrological processes of two watersheds in the Greater Bay Area (GBA), a critical economic zone in China that encompasses major cities such as Guangzhou, Shenzhen, Hong Kong, and Macao. According to official data, the GDP of the GBA exceeded 14 trillion yuan in 2023 (Greater Bay Area, 2024). Despite this economic success, the region faces significant challenges in achieving sustainable growth under rapid urbanization, making it an ideal case study for investigating the impacts of development on hydrological processes. For this study, we use the Simulator for Hydrologic Unstructured Domains (SHUD) as an ISSHM. It examines the influences of terrain slope and urbanization-driven LULCC on the hydrological components of surface runoff, subsurface flow, ET, and infiltration at both daily and annual scales.

## 2 Study site

The study focuses on two neighboring watersheds within the Shenzhen River and Bay Basin (SRBB) in the GBA—the Ng Tung River Watershed (NTRW) in Hong Kong and the Buji River Watershed (BJRW) in Shenzhen (Figs. 1a and 1b). The NTRW encompasses an area of 70.7 km², while the BJRW covers 66.3 km². Situated in a subtropical region, the SRBB experiences an average annual temperature of 23.3°C and receives a substantial amount of precipitation, averaging 1933 mm annually, with significant inter-annual variability. Notably, about 86% of this precipitation falls during the monsoon season (April–September), with the region experiencing an average of 130 rainy days per year. The intensity of daily rainfall during this period can be significant, reaching 289 mm and 382 mm for the 10-year and 50-year return period events, respectively.

Despite their proximity and similar climatic conditions, the NTRW and BJRW exhibit distinct differences in topography and land use patterns. The NTRW is characterized by steep slopes, with an average gradient of 12.3° and elevation variations ranging from 0.5 to 611.6 m (average elevation 97.1 m). In contrast, the BJRW features relatively flatter terrain, with an average slope of 7.5° and elevation ranging from 0.5 to 435.3 m (average elevation 70.6 m) (Fig. 1c). These watersheds demonstrate the rapid urbanization of Shenzhen and Hong Kong since the 1980s; however, urbanization has progressed more rapidly in the BJRW. Initially, the BJRW had limited construction areas with forests predominating (Cheng et al., 2023). By 2020, built-up land in the BJRW had increased to 71%, while in the NTRW, forests remain dominant and built-up areas constitute 37% of the land (Fig. 1d).

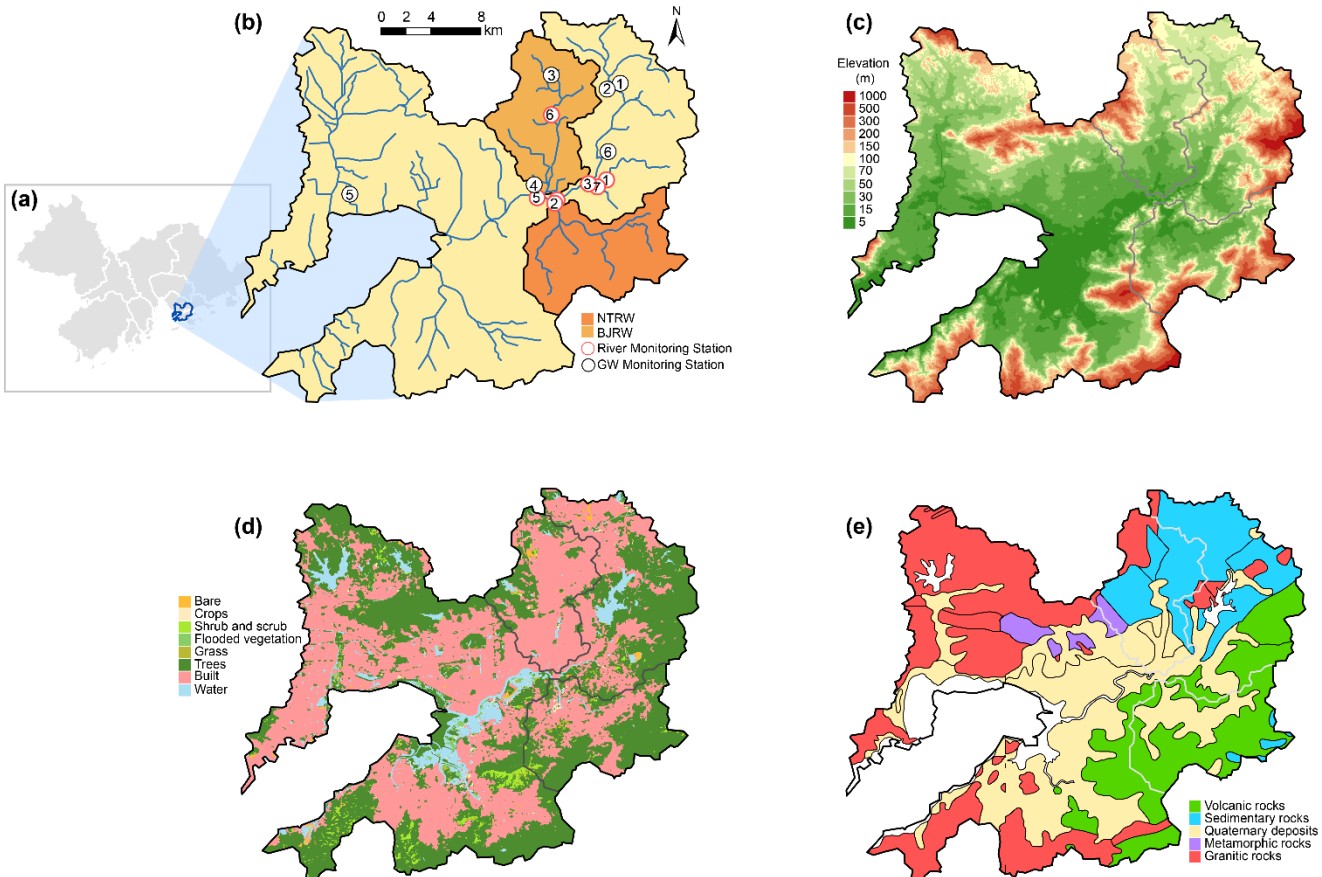

**Figure 1.** Location and characteristics of the Ng Tung River Watershed (NTRW) and Buji River Watershed (BJRW): (a) location of the Shenzhen River and Bay Basin (SRBB) within the Greater Bay Area (GBA), (b) location of the NTRW (dark orange) and BJRW (light orange) within the SRBB (yellow), along with channels (blue), calibration river monitoring stations (numbered 1−7, red circles), and calibration groundwater monitoring stations (numbered 1−6, black circles), (c) DEM (FABDEM V1-2), (d) land cover map of 2020, (e) geological map.

## 3 Methodology

### 3.1 Hydrological model

The hydrological model employed in this study is SHUD (Shu et al., 2020), which evolved from the well-known Penn State Integrated Hydrologic Model (PIHM; Qu and Duffy, 2007; Kumar, 2009; Kumar et al., 2009). SHUD is an open-source model that incorporates a user-friendly data preprocessing toolkit, rSHUD (Shu et al., 2024), designed to simplify tasks such as grid partitioning, data integration, and model setup, addressing common challenges faced by hydrologists when working with ISSHMs. By integrating the parallel programming framework OpenMP, SHUD achieves high computational efficiency and has demonstrated superior robustness in solving problems at the watershed scale compared to PIHM, thus confirming its effectiveness in hydrological modeling (Shu et al., 2020).

As illustrated in Fig. 2, the hydrological processes simulated by SHUD include rainfall, surface water ponding storage,

surface water infiltration, surface runoff, ET, changes in unsaturated layer moisture, groundwater flow, and river flow processes.

The model represents the land domain using unstructured triangular elements and trapezoid segments for the river network.

Each triangular element is vertically discretized into three layers: the top layer represents the land surface, the middle layer

represents the unsaturated zone, and the bottom layer represents the saturated aquifer. The model employs the finite volume

method to spatially discretize the partial differential equations of hydrological states into ordinary differential equations,

enabling detailed simulation of hydrological dynamics.

For a more comprehensive understanding of the four hydrological processes analyzed in this study, we provide the

relevant assumptions and computational formulas used in SHUD in Appendix A. Further details on the mathematical and

algorithmic structure of SHUD are available in the referenced papers (Shu et al., 2020; Shu et al., 2024) and on the SHUD

Book website (SHUD Book, 2024).

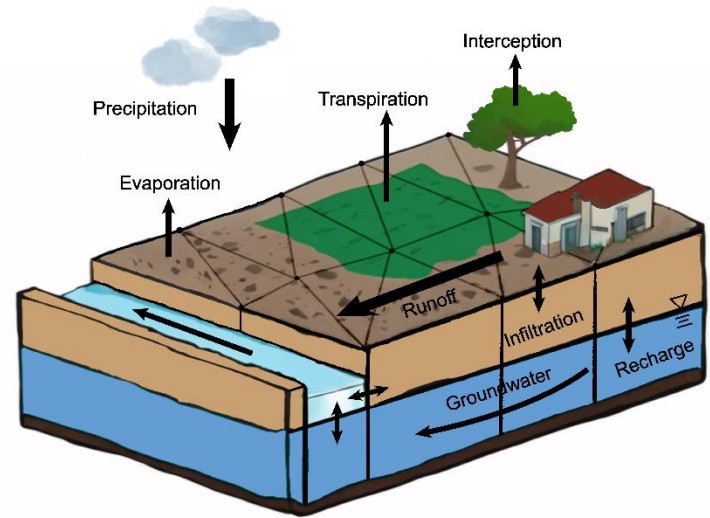

**Figure 2.** Model schematic of hydrological processes in the SHUD model.

**3.2 Data collection and model setup**

We set up the model domain as the entire SRBB, rather than focusing solely on its smaller two watersheds. This decision was

driven by two strategic considerations. Firstly, the limited availability of monitoring data within the two watersheds

necessitated a broader spatial framework to ensure a comprehensive dataset for robust hydrological analysis. Secondly, the

similar characteristics of geology (Fig. 1e), soil (Fig. 3d), and vegetation (Fig. 3e) across the SRBB and its subbasins supported

the feasibility of this extensive modeling approach. The SRBB, covering an area of 596 km², was discretized into 6,602

triangular meshes. Specifically, the NTRW and the BJRW were represented by 819 and 793 triangular grids, respectively (Fig.

3a). In the model, the outer boundary of the SRBB was designated as a zero-flow boundary, meaning no water flows across

this boundary. Additionally, the land and river boundaries along the concave boundary in the southwestern part of the basin

were set as a fixed head value, corresponding to the local sea level. This fixed-head boundary was established at 1.5 m, based

on annual tidal observations from the Hong Kong Observatory (HKO). While this fixed-head approximation does not account

for the precise daily tidal fluctuations, it represents a reasonable compromise for hydrological modeling purposes. Given that

the two watersheds are situated significantly inland from the ocean, their hydrological processes are minimally affected by

tidal variations.

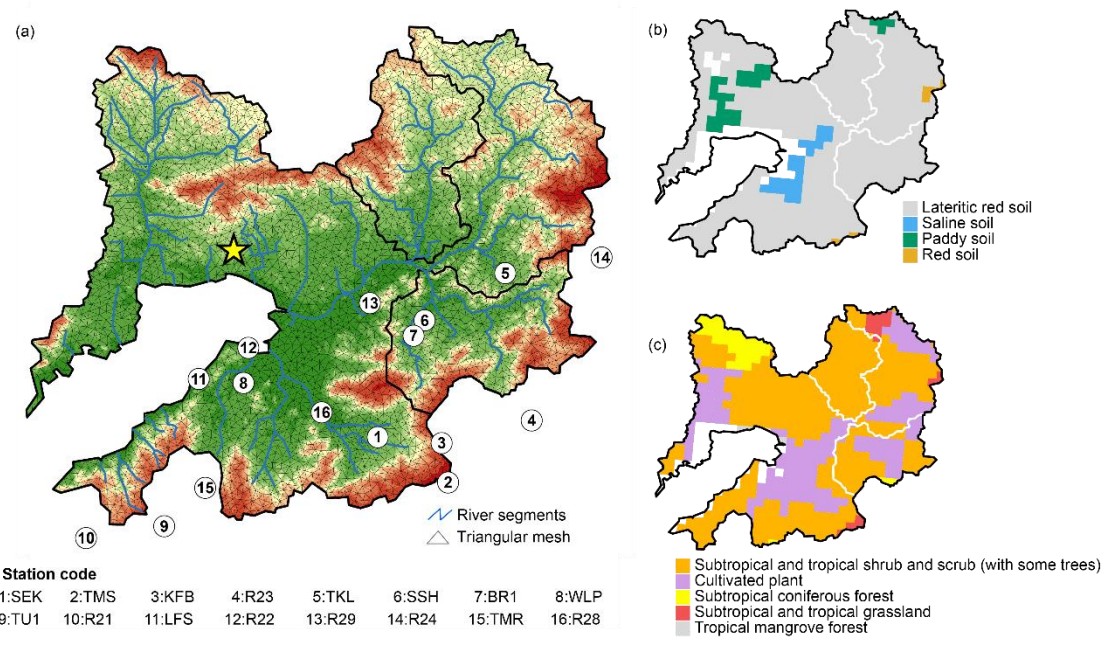

**Figure 3.** Map of meteorological site locations and triangular meshes of two watersheds. Black circles (numbered 1-16) represent rainfall sites located in Hong Kong, and the yellow star represents the Shenzhen Meteorological Station (SMS) (a), soil map (b), and vegetation map (c).

The Digital Elevation Model (DEM) for the study area was sourced from the FABDEM V1-2 dataset (Neal and Hawker, 2023) and offers a resolution of 30 meters. Land cover data for 2020, with a spatial resolution of 10 meters, were acquired from the Dynamic World Project via Google Earth Engine (Brown et al., 2022). Data on soil types and vegetation were obtained from the Data Center for Resources and Environmental Sciences at the Chinese Academy of Sciences (RESDC, 2024), and geological information was sourced from the China Geological Survey (GeoCloud, 2024). Satellite imagery was utilized to determine river channel widths. Determining the appropriate soil depth remains a significant challenge, and as highlighted by Fan et al. (2019), weathering fractures notably influence hydrological activities. Based on the geological data from the study site, extensive weathering is noted in the mountainous regions. Consequently, the aquifer depth was modeled to vary gradually from 18 meters in the upslope areas to 9 meters downstream.

Additionally, driving force data were collected for two distinct periods. The first period, from 2020 to 2021, included hourly meteorological data from the Shenzhen Meteorological Station (SMS), provided by the Meteorological Bureau of the

Shenzhen Municipality. This dataset included records of precipitation, temperature, relative humidity, and wind speed. Hourly precipitation data for the same period were also gathered from 16 additional gauging sites in Hong Kong, sourced from the HKO (Fig. 2a). The second period, from 1993 to 2021, involved collecting precipitation data from the R29 station via the HKO. Moreover, monitoring data of daily river discharge from seven stations and daily or weekly groundwater table depths from six stations were gathered from the Water Authority of the Shenzhen Municipality for the period of 2020–2021. While urban drainage could affect river discharge in Shenzhen, river rehabilitation projects through 2020 (Buji Sub-district Office, 2024) helped minimize drainage network inflows. Therefore, we assume the monitored river discharge data collected during 2020–2021 can be fully attributed to terrestrial runoff intercepted along the river channels. A comprehensive summary of all datasets and related information is provided in Table 1.

**Table 1.** Summary of collected datasets and related information.

| Data | Source | Resolution | Time period | Purpose |
|---|---|---|---|---|
| DEM | FABDEM V1-2 | 30 m | | |
| Land cover type | Dynamic World Project | 10 m | 2020 | |
| Soil type | RESDC | 1000 m | | Model mesh grid attributes set up |
| Geology | China Geological Survey (GeoCloud) | 100 m | | |
| River characteristics | Google Earth | | | |
| Meteorological data of the Shenzhen Meteorological Station (SMS) | Meteorological Bureau of the Shenzhen Municipality | Hourly | 2020–2021 | (1) Model calibration phase driving force inputs; (2) Model scenarios 1 and 2 driving force inputs |
| Precipitation of 16 Hong Kong stations | Hong Kong Observatory (HKO) | Hourly | 2020–2021 | Model calibration phase driving force inputs |
| Precipitation of the R29 station | Hong Kong Observatory (HKO) | Hourly | 1993–2021 | Model scenarios 3 and 4 driving force inputs |
| Streamflow monitoring data of 7 sites | Water Authority of the Shenzhen Municipality | Daily | 2020–2021 | Model calibration |
| Groundwater table depth monitoring data of 6 sites | Water Authority of the Shenzhen Municipality | Daily or weekly | 2020–2021 | |

**3.3 Model calibration**

We employed rainfall data from 17 sites covering the period from 2020 to 2021 to drive the model during the calibration process. To distribute the rainfall data effectively across all 17 sites, we utilized the Thiessen multi-polygon method, allocating the data to corresponding triangular grids. Due to limitations in data availability, meteorological parameters such as temperature, relative humidity, and wind speed were sourced solely from the SMS for the entire basin. The initial setup of the

model parameters was informed by field data, the general features of the model structure, and past modeling experience. The model underwent multiple spin-up sessions using 2020 meteorological data to establish an initial condition that closely mirrors the monitoring datasets.

Given the heterogeneity of the basin and the calibration target covering two types and multiple sites of monitoring datasets, effective automatic calibration becomes extremely difficult. Therefore, manual calibration methods are often preferred for ISSHMs (Shi et al., 2014; Thornton et al., 2022; Brandhorst and Neuweiler, 2023). Monitoring data from the entire period were utilized for calibration, focusing on enhancing model performance. Parameter selection was guided by prior ISSHM calibration experience, insights from the literature (Baroni et al., 2010; Song et al., 2015; Liu et al., 2020), and preliminary sensitivity analyses. Informed by these combined efforts, we identified seven critical parameters related to unsaturated zone and aquifer properties for calibration (Table 3).

As the calibrated parameters were not independent, an iterative adjustment process was required. Initially, all parameters were coarsely adjusted to match the simulation river flow with monitoring data, emphasizing trends, peak timing, and peak values, even though consistency in baseflow simulation results was not yet achieved. The next stage focused primarily on modifying aquifer-related parameters to ensure that the simulated baseflow closely matched the monitoring results. In the final stage, the groundwater table was calibrated by refining soil and aquifer parameters near the monitoring sites while minimizing significant changes to previously established parameters. These three stages were repeated until the model met our performance criteria, defined as achieving a Nash-Sutcliffe Efficiency (NSE) for streamflow greater than 0.5 and simulated groundwater tables falling within acceptable observational ranges. A detailed discussion of the final calibrated parameters and results is provided in Sect. 4.1.

**3.4 Scenario design and evaluation methods**

We developed four modeling scenarios differentiated by time span and land use pattern (Table 2). Scenarios 1 and 2 analyze hydrological processes at daily and annual temporal resolutions, respectively, using continuous meteorological data provided by the SMS for the years 2020–2021. These scenarios aim to determine how watershed slope and urbanization conditions influence daily and annual hydrological responses. Scenarios 3 and 4 extend the analysis to a 29-year period (1993–2021), utilizing rainfall data from the R29 station. These scenarios enrich our understanding of how annual rainfall variability influences topographic slope and LULCC on hydrological processes. The overall framework of our assessment methods is illustrated in Fig. 4, with detailed descriptions of land use pattern settings and statistical methods provided in Sects 3.4.1 and 3.4.2, respectively.

**Table 2.** Designed four scenarios.

| Scenario | Driving force inputs time span | Land use pattern |
|---|---|---|
| 1 | 2020–2021 | HLU |
| 2 | 2020–2021 | CLU |
| 3 | 1993–2021 | HLU |
| 4 | 1993–2021 | CLU |

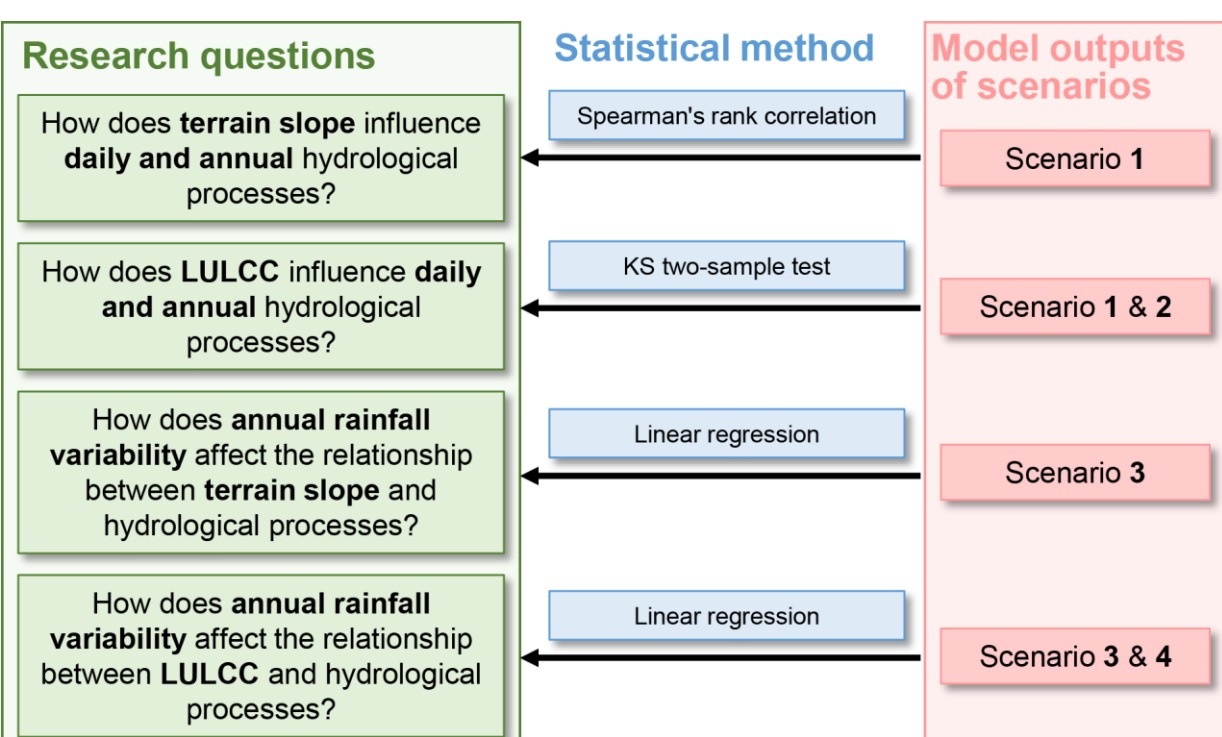

**Figure 4.** Framework for assessing the impacts of slope and LULCC on hydrological processes.

**3.4.1 Two land use patterns**

Among the four scenarios, we implemented two types of land use patterns: Current Land Use (CLU) and Historical Land Use (HLU). The CLU pattern was derived from 2020 land use data, which was obtained from the Dynamic World project, with a spatial resolution of 10 meters (Fig. 1d). The CLU pattern was generated by determining the dominant land use type based on areal coverage for each triangular mesh grid and assigning that classification to the corresponding grid (Fig. 5a). To generate the HLU pattern, we modified the CLU pattern by reclassifying all mesh grids identified as built-up land to tree cover in both watersheds, simulating pre-urbanization conditions (Fig. 5b).

Both the original raster data and our hydrological model incorporate eight land use classifications: bare land, crops, shrubs and scrubs, grassland, flooded vegetation, trees, built-up land, and water bodies (Fig. 1d and Fig. 5). Each land use type is parameterized with specific values in the model, including leaf area index (LAI), albedo, surface roughness, root zone depth, and impervious surface fraction. The impervious surface fraction is set to 94% for built-up land, as these areas represent high-density urban development. All other land use types are assigned an impervious surface fraction of 0%. Under the CLU pattern,

built-up land comprises 37.6% of the NTRW and 69.8% of the BJRW. Following reclassification in the HLU pattern, the built-up land fraction becomes 0% in both watersheds.

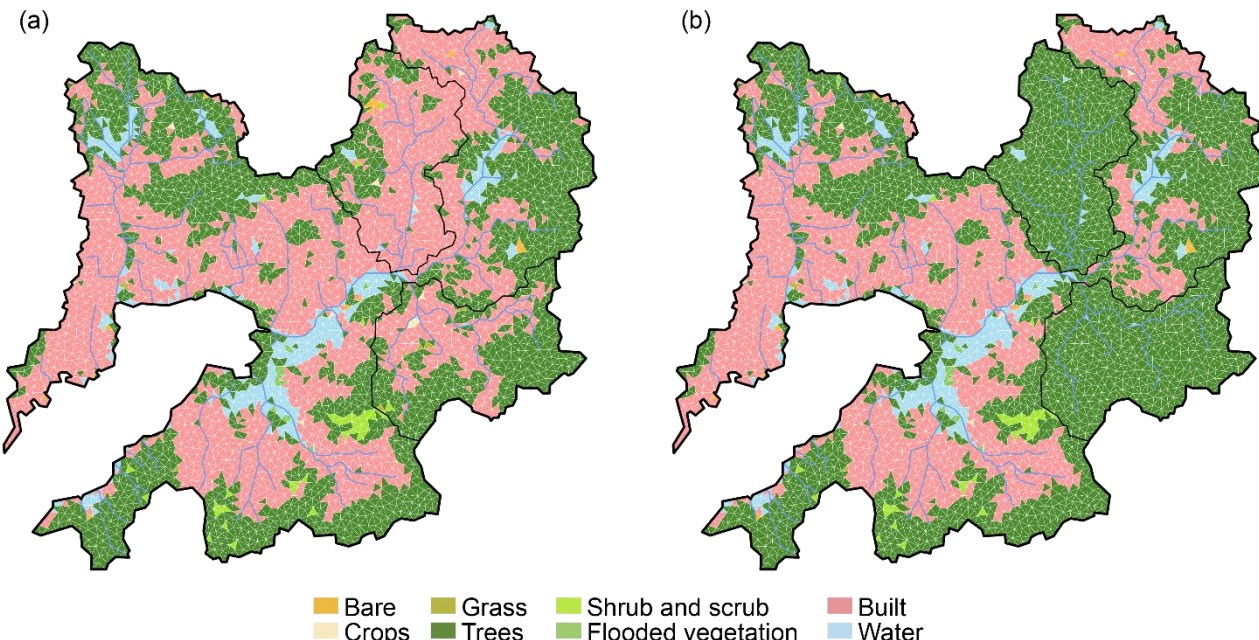

**Figure 5.** Model setup of land use patterns for two watersheds: (a) Current Land Use (CLU) pattern showing the present urbanized state with extensive built-up areas (pink) mixed with other land cover types, and (b) Historical Land Use (HLU) pattern representing pre-urbanization conditions, where all built-up areas have been converted back to trees (dark green) to simulate the historical natural state.

**3.4.2 Assessment of slope and LULCC effects**

To isolate the impact of slope from LULCC effects, we analyzed slope impacts within the two watersheds exclusively under the HLU pattern. To ensure a coherent assessment of how slope influences hydrological processes, we derived slope values based on the topographical characteristics of the model instead of the original 30-meter resolution DEM data. We extracted elevation values for each triangular mesh vertex from the original 30-meter DEM data, re-interpolated these values to create a new raster DEM, and then calculated the average slope for each mesh grid.

For a more detailed examination of slope impacts across different spatial areas within the watersheds, we divided the watersheds into three elevation zones. First, we calculated the average elevation of each triangular mesh grid. Using the natural breaks method, we classified all grids into six elevation groups, with the first and second natural breakpoints at approximately 40 m and 120 m. To ensure sufficient grids for reliable statistical analysis, we grouped the remaining four elevation categories into a single elevation zone. Based on these criteria, we defined three elevation zones:

- **Zone 1** consists of low-elevation grids with DEM values below 40 m, primarily flat regions.
- **Zone 2** includes grids with DEM values from 40 m to 120 m, located at mountain foothills.
- **Zone 3** comprises high-elevation grids with DEM values above 120 m.

After classification, the mean slope values for each zone are shown in Fig. 6. Since the NTRW terrain is generally steeper,

the average slope value for each zone is greater in NTRW than in BJRW.

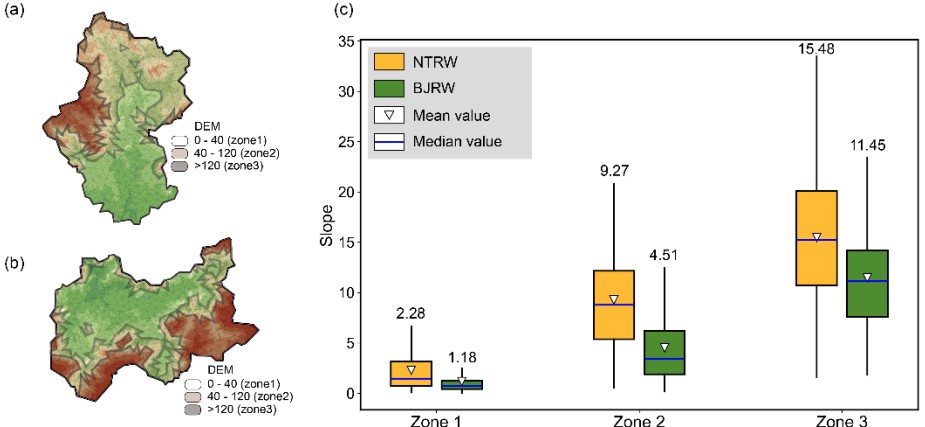

**Figure 6.** Elevation-based delineation of three zones in BJRW (a) and NTRW (b), classified using DEM data as Zone 1 (0-40 m), Zone 2

(40-120 m), and Zone 3 (>120 m). Statistical distribution of slopes within these zones illustrated through box plots (c), with mean values

labeled numerically.

The statistical method used to examine the influence of slope on hydrological responses is Spearman's rank correlation

method (Seibert et al., 2003; Hauke and Kossowski, 2011). To analyze how annual rainfall variability affects the correlation

between topographic slope and hydrological processes, we developed a simple linear regression model (Appendix B).

To evaluate the impacts of LULCC, we compared hydrological outputs between CLU and HLU patterns. We employed

the Kolmogorov-Smirnov (KS) two-sample test (Lilliefors, 1967) to assess the statistical significance of LULCC-induced

changes in hydrological responses. To investigate how annual rainfall variability influences the relationship between LULCC

and hydrological processes, we developed a simple linear regression model, following the slope assessment method (detailed

in Appendix B).

**4 Results and discussion**

**4.1. Model performance**

Due to spatial heterogeneity within the watersheds, the calibrated values for each parameter are formed as a matrix. For clarity,

only the median values is displayed (Table 3). The first four parameters, $K_s$, $\theta_{ss}$, $\alpha$ and $\beta$, are primarily associated with the

vadose zone and significantly influence the hydraulic processes in the soil layer. The last three parameters $K_g$, $\theta_{gs}$ and $\theta_{gr}$,

govern the hydraulic processes in the aquifer layer. All these parameters fall within reasonable ranges, as supported by previous

studies (Das, 1990; Freeze and Cherry, 1979; Bear, 2013; Van Genuchten, 1980).

Figures 7a–c display the hydrographs of daily simulated and observed streamflow at various river gaging stations within the BJRW (Site 6; Fig. 7c), at the upstream of the watersheds (Site 1; Fig. 7a), and at the downstream of the watersheds (Site 2; Fig. 7b), respectively. The NSE indices, computed for the entire simulation period, demonstrate satisfactory model performance, except for Site 2 where the observed dataset shows daily fluctuations in river flow during rain-free periods due to tidal influences. Therefore, for such sites, we specifically calibrated the discharge during rainy days and calculated the NSE index using data from those days. The simulation results exhibit satisfactory performance with NSE indices greater than 0.5, indicating a reasonable accuracy in streamflow predictions.

Furthermore, the monthly calibration results reinforce the robust performance of the calibrated model, exhibiting $R^2$ values exceeding 0.6 (Figs. 7d–f; Moriasi et al., 2007). This strong correlation suggests a consistent and reliable model behavior over a longer time scale. Figures 7g–i present the comparisons between the simulated and observed groundwater data. It is challenging to evaluate the assessment indices of groundwater calibration for such long durations. However, our calibration outcomes indicate a marked concordance between the model outputs and observed data trends, and the modeled groundwater table depth closely aligns with the measured depths, underscoring the model's accuracy in reflecting actual groundwater conditions. Overall, the model exhibits satisfactory performance on both surface and subsurface water flows. Additional sites' calibration results are available in Fig. C1.

**Table 3.** Refined parameters for the watershed after calibration.

| Parameter | Description | Allowable value range | Median value after calibration | Unit |
|---|---|---|---|---|
| $K_s$ | Soil saturated infiltration conductivity | $10^{-3}$–$10^4$ | 0.045 | m day$^{-1}$ |
| $\theta_{ss}$ | Soil saturated water content | 0.25–0.7 | 0.531 | - |
| $\alpha$ | van Genuchten parameter | >0 | 5.23 | m$^{-1}$ |
| $\beta$ | van Genuchten parameter | >1 | 1.29 | - |
| $K_g$ | Groundwater hydraulic conductivity | $10^{-5}$–$10^4$ | 2.6 | m day$^{-1}$ |
| $\theta_{gs}$ | Groundwater saturated water content | 0.0–0.5 | 0.3 | - |
| $\theta_{gr}$ | Groundwater residual water content | 0.0–0.5 | 0.01 | - |

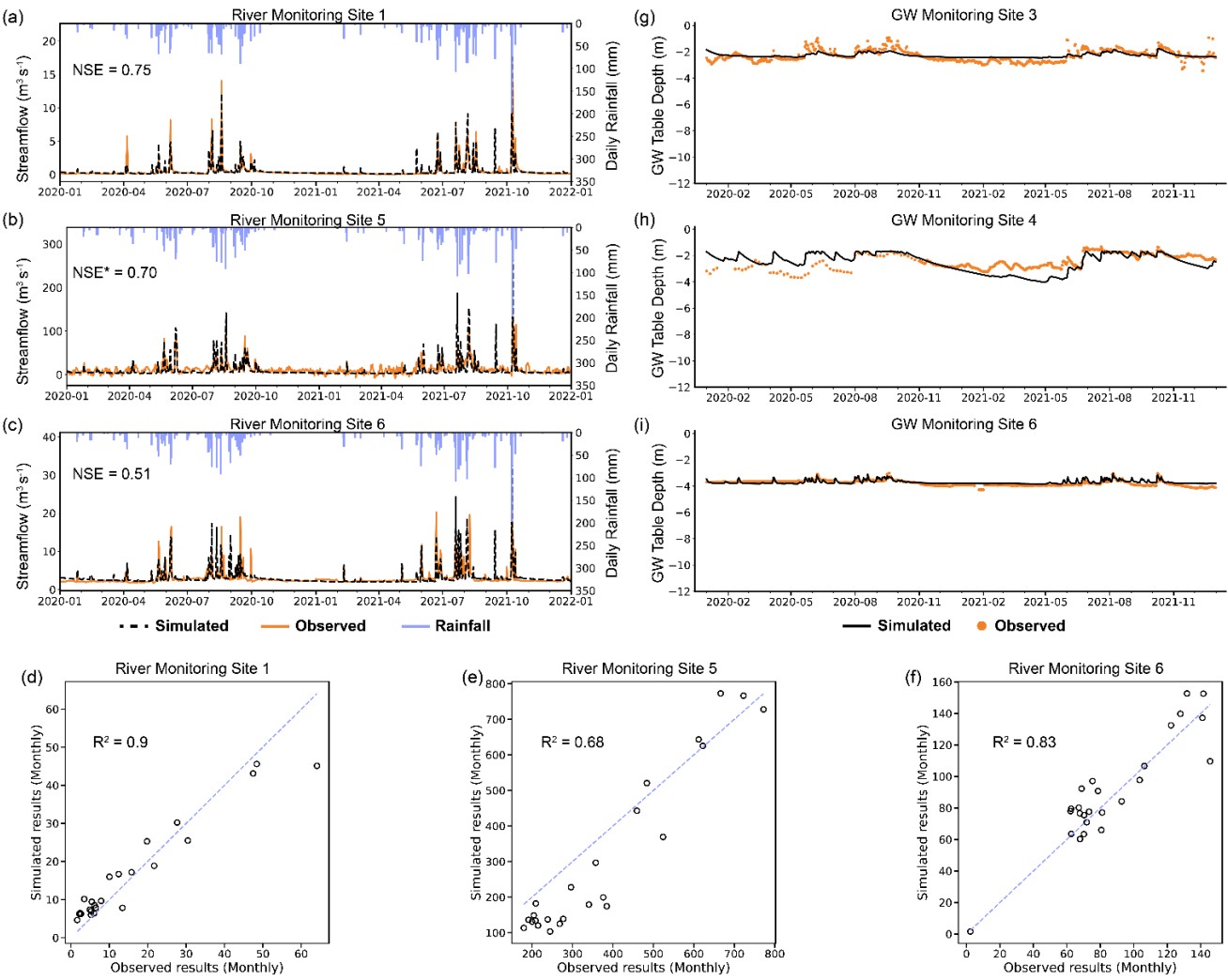

**Figure 7.** Calibration performance of SHUD model across daily river discharge in the river monitoring sites 1, 5 and 6 (a)–(c), and monthly river discharge in the river monitoring sites 1, 5 and 6 (d)–(f), and groundwater table depth in the groundwater monitoring sites 3, 4 and 6 (g)–(i).

**4.2 Daily and annual scale hydrological responses**

**4.2.1 Stronger correlation between slope and daily subsurface flow**

Figure 8 depicts the Spearman correlation test between four hydrological processes and terrain slope on a daily scale (i.e., on the rainy days), with all depicted markers being statistically significant (p-value ≤ 0.05). The analysis primarily emphasizes slope, but also explores the influence of daily rainfall to provide additional insights. The correlation analysis between daily rainfall and hydrological processes reveals distinct patterns of influence. Infiltration and surface runoff demonstrate the strongest response to rainfall amounts, with correlation coefficients ranging from -0.6 to 1, while their correlation with terrain slope remains relatively weak (between -0.2 and 0.2) in all zones of the two watersheds. ET emerges as the third most strongly correlated process with rainfall. Notably, subsurface flow exhibits a different pattern, showing a stronger correlation with local slope (coefficients between -0.4 and 0.2) than with rainfall amounts (coefficients between -0.2 and 0.2) during rainy days. This

finding aligns with existing literature, highlighting the critical role of topography in influencing groundwater dynamics during rainfall events (Hopp and McDonnell, 2009; Detty and McGuire, 2010; Jencso and McGlynn, 2011; Singh et al., 2021). In both watersheds, the relationship between slope and subsurface flow varies with elevation, revealing a complex interplay between topography and groundwater dynamics. A negative correlation exists between slope and subsurface flow in Zones 2 and 3, while a positive correlation is observed in Zone 1. This indicates that in the low-elevation Zone 1, as slope increases, subsurface outflow also increases, while in the mid- and high-elevation Zones 2 and 3, as slope increases, subsurface flow decreases. In low-elevation areas, the groundwater table is typically shallow and the soil is relatively saturated. Under these conditions, increasing slope significantly enhances the lateral hydraulic gradient, thereby facilitating downslope groundwater flow. In mid- to high-elevation areas, the groundwater table is generally deeper. Steeper slopes tend to boost surface runoff, reducing infiltration and diminishing groundwater recharge. Consequently, a negative correlation arises between slope and groundwater outflow in these higher elevation zones.

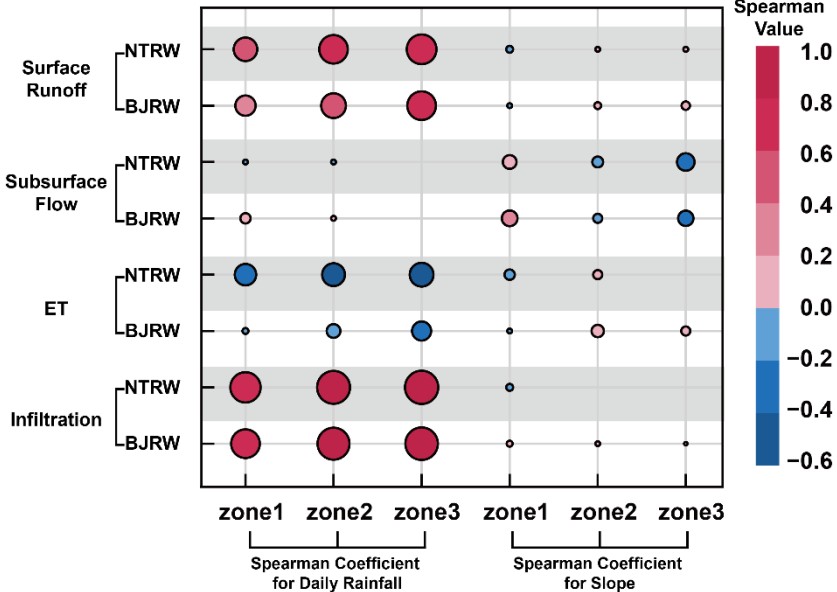

**Figure 8.** Comparative analysis of slope influence and daily rainfall on four hydrological variables. Marker size denotes the absolute value of the Spearman correlation coefficients, while marker color indicates the direction of the relationship between slope or rainfall and the four model outputs. Generally, red represents a positive correlation, whereas blue denotes a negative correlation.

**4.2.2 Faint slope-flow relationship in NTRW's lower zone**

Figure 9 presents the comparative results of terrain slope at daily and annual scales. The findings suggest that slope has a more pronounced relationship with annual surface runoff, subsurface flow, and infiltration at higher elevations (Zone 3) compared to daily scales. This pattern emphasizes the pivotal role of slope in redistributing water post-rainfall events in mountainous regions. Seibert et al. (2003) and Rinderer et al. (2014) noted that topographic indices more accurately reflect hydrological responses under steady-state conditions. Specifically, Rinderer et al. (2014) reported from their analysis of data from 51

groundwater wells in a Swiss catchment that the ability of the TWI to predict water table distributions diminishes under unsteady conditions. These findings from previous studies align with our results, where the stronger correlations observed at annual (more steady-state) scales compared to daily (unsteady) scales suggest that topographic controls on hydrological processes are more pronounced and predictable over longer time periods when the system approaches steady-state conditions.

In mid-elevation regions (Zone 2), the most significant finding is the positive correlation between annual ET and local slope. This relationship suggests that steeper slopes in mid-elevation zones exhibit higher annual ET amounts. Spearman correlation analysis (results not shown) between slope and annual average soil moisture across Zone 2 grids revealed a correlation coefficient of 0.25 (p-value < 0.05), indicating a positive correlation. Areas with steeper slopes have higher soil moisture, potentially contributing to higher ET amounts. Lee and Kim (2022) reported similar findings in the Sulmachun watershed, Korea, where they observed a positive correlation between surface (10 cm) soil moisture and surface slope through April-December monitoring.

Analysis of annual flow processes at lower elevations (Zone 1) reveals a strong correlation between terrain slope and hydrological behavior in the gently sloping BJRW. However, this correlation is markedly weak in the steeper NTRW. This difference can be explained by the rapid water movement in steeper watersheds (Fan et al., 2019; Singh et al., 2021), where hydrological processes at lower elevations are dominated by swift upstream inflows rather than local topographic features. Conversely, watersheds with gentler slopes experience slower flow processes, allowing local topography at lower elevations to persistently influence water flow pathways.

The comparison between daily and annual scales reveals distinct temporal characteristics in slope and hydrological process relationships. At the daily scale, surface processes show immediate responses to rainfall with weak slope correlations, while subsurface flow exhibits stronger topographic control. However, at the annual scale, the influence of slope becomes more pronounced across all hydrological processes, particularly in higher elevations. This scale-dependent behavior suggests that while local topography may have limited immediate impact on daily hydrological processes, its cumulative effects become increasingly significant over longer time periods. This temporal distinction is particularly evident in watersheds with different slope gradients. In steep watersheds, lower-elevation regions show weak correlation with local slope, while in watersheds with gentle slopes, local topographic features have a more persistent influence on flow pathways. These findings highlight the importance of considering both temporal scales and watershed characteristics in understanding topographic controls on hydrological processes.

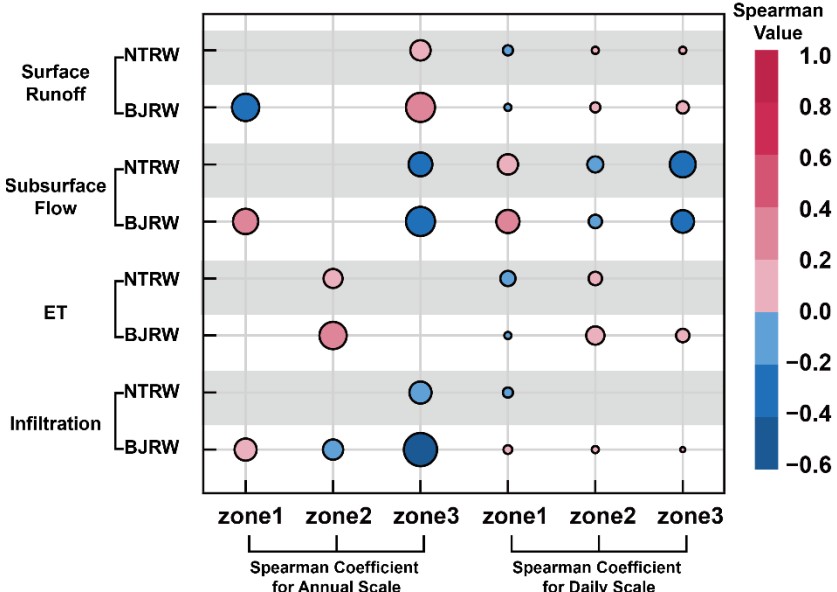

**Figure 9.** Comparison of hydrological responses to slope variability on annual and daily scales in NTRW and BJRW.

**4.2.3 Dominant impact of LULCC on daily infiltration**

Figure 10a illustrates the absolute mean differences in rainy-day hydrological outputs between the HLU and CLU patterns for each grid cell. Employing the KS statistic test, significant alterations in the cumulative distribution function (CDF) of daily hydrologic outputs were identified, highlighting the substantial impacts of LULCC. Among the hydrological processes examined, daily infiltration exhibits the most pronounced and widespread differences, underscoring the dominant influence of LULCC. When considering only absolute mean differences, surface runoff is identified as the second most influenced processes. This finding aligns with the results of Chu et al. (2010) and Diem et al. (2021), which underscore the extensive impact of urbanization on surface runoff through changes in infiltration.

Regions with a KS statistic greater than 0.5 are considered to be significantly affected by urbanization. The spatial statistical characteristics of these regions for four hydrological processes are analyzed in Figs. 10b–d. Infiltration exhibits the most extensive spatial impact, whereas changes in surface runoff, subsurface flow, and ET are confined to more limited areas (Fig. 10b). Considering the elevation variations, the influenced surface runoff and ET regions are more significant at higher elevations, while the most influenced subsurface flows are limited to lower elevation regions (Fig. 10c). Notably, areas with significant ET changes are characterized by steeper slopes (Fig. 10d). Figure 10 demonstrates that hydrological processes most influenced by urbanization are not uniform but rather concentrated in specific regions.

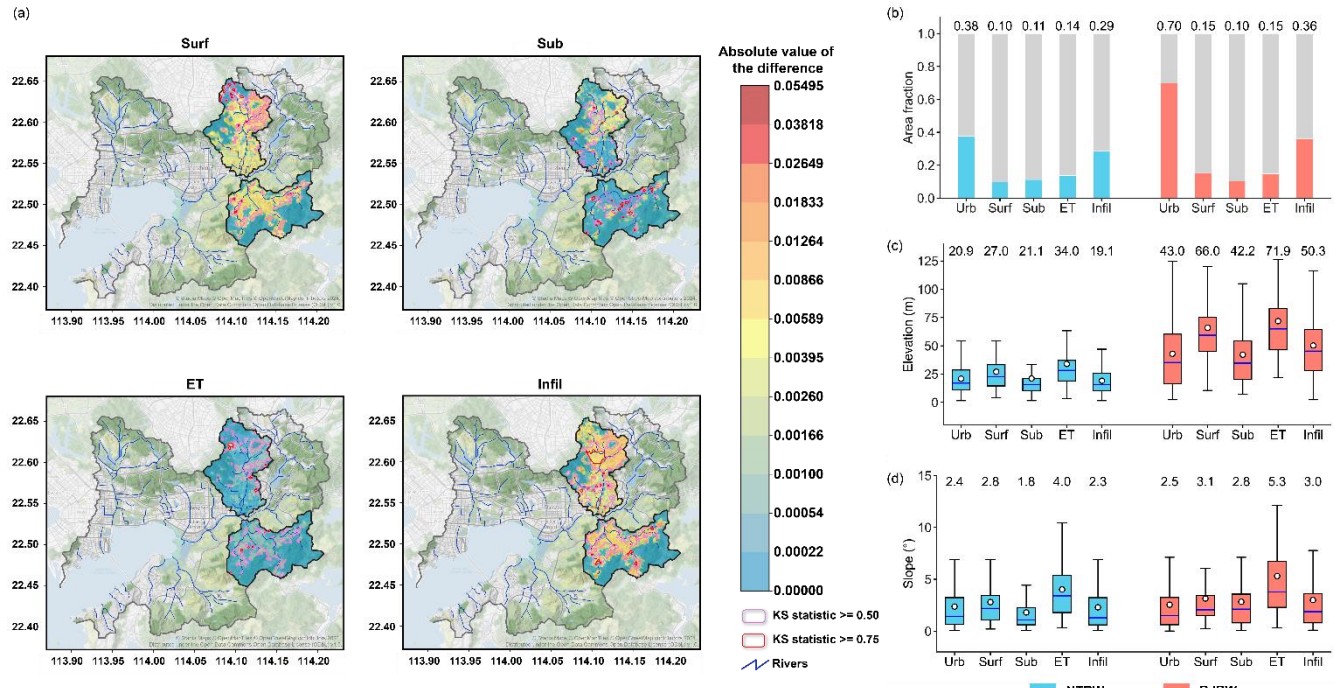

**Figure 10.** Spatial analysis of urbanization impacts on hydrological processes in NTRW and BJRW. (a) Spatial distribution of KS statistics and absolute differences between first and second scenarios for four hydrological processes: surface runoff (Surf), subsurface flow (Sub), evapotranspiration (ET), and infiltration (Infil). The color scale represents the absolute value of differences, with areas outlined in pink and red indicating KS statistics > 0.5 and > 0.75, respectively. (b) Percentage of significantly affected areas (KS > 0.5) for each hydrological process. (c) Elevation distribution and (d) slope distribution of significantly affected areas, with blue and red boxes representing NTRW and BJRW, respectively. Box plots show the median (horizontal line), 25th and 75th percentiles (box boundaries), and the mean value (white dot with corresponding text above each box).

### 4.2.4 NTRW shows more sensitivity to LULCC

The KS test indicates statistically significant changes in all four hydrological outputs at an annual scale after urbanization, with all p-values below 0.05 (Fig. 11). The results depict an increase in annual surface runoff and reductions in subsurface flow, ET, and infiltration following urbanization. This aligns with findings from Shao et al. (2020), who used a process-based hydrological model to examine the response of surface runoff to LULCC in two adjacent watersheds in Texas, USA. They reported that urbanization leads to increased runoff, a finding consistent with our results. Furthermore, the KS test results reveal relative consistency within each watershed for surface runoff, ET, and infiltration values. Specifically, in the NTRW, the KS values for surface runoff, ET, and infiltration are recorded at 0.39, 0.395, and 0.377, respectively. The corresponding values in the BJRW are 0.531, 0.583, and 0.615. However, subsurface flow shows lower KS values of 0.127 in the NTRW and 0.263 in the BJRW, suggesting that urbanization has a less impact on the annual subsurface flow process.

Although urbanized land accounts for 69.8% of the land cover change in the BJRW, resulting in more pronounced responses in the four hydrological processes compared to the NTRW (where urbanized land comprises only 37.6% of the change), it is noteworthy that per unit of urbanized area, the flatter watershed demonstrates a greater capacity to mitigate the

effects of LULCC. This is evidenced by the KS values for surface hydrological processes in the BJRW (ranging from 0.531 to 0.615) being lower than the proportion of urbanized land change (0.698). In the NTRW, the KS values for surface hydrological processes (ranging from 0.377 to 0.395) are slightly higher than the proportion of urbanized land change (0.376). This observation is supported by Zhou et al. (2015), who noted that flatter terrains tend to absorb changes more effectively due to prolonged water-soil contact times, which enhance infiltration and storage capacities. This capacity may help mitigate the more severe hydrological alterations typically associated with extensive urbanization.

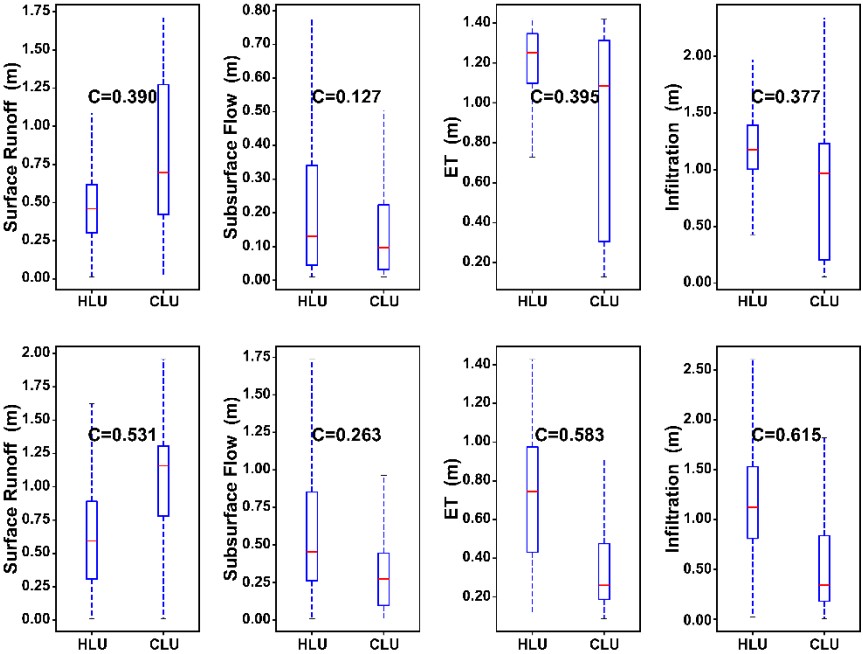

**Figure 11.** Box plots delineating the impacts of LULCC on the four annual outputs across all meshes within each watershed. The comparison contrasts the outcomes under the HLU and CLU patterns. The top row displays the results of NTRW, while the second row displays the results of BJRW. KS test values (C) are annotated, all p-values are less than 0.05.

**4.3 Variations with different annual rainfall amounts**

**4.3.1 Rainfall intensifies subsurface flow-slope relationship in BJRW's lower zone**

Figure 12 presents scatterplots and regression equations that analyze the correlation between annual precipitation and Spearman statistic values from 1993 to 2021, highlighting outcomes that are statistically significant (p-value ≤ 0.05), as identified in Sect. 4.2.2. The analysis shows minimal changes in Spearman statistic values across most study areas; however, a notable variation was observed in subsurface flow within Zone 1 of the BJRW, where a coefficient of 0.07 indicates that each 100 mm increase in annual precipitation enhances the correlation between slope and subsurface flow by 0.007. This change corresponds to a shift in the Spearman coefficient from 0.174 to 0.258 as annual rainfall increases from 1200 mm to 2400 mm. This observation is supported by findings from Zhang et al. (2022a), who reported that under scenarios of higher precipitation

and greater hydraulic conductivity, the extent and permeation depth of the saturated zones beneath mountains exhibit a stronger

correlation with the terrain. This effect is likely due to increased precipitation levels raising the water table at lower elevations,

thus enhancing the relationship between slope and subsurface flow.

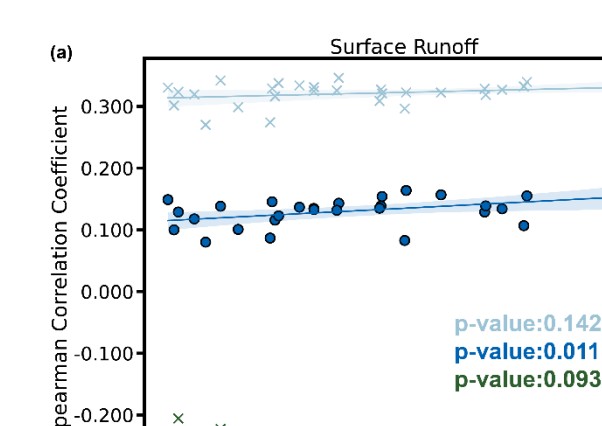
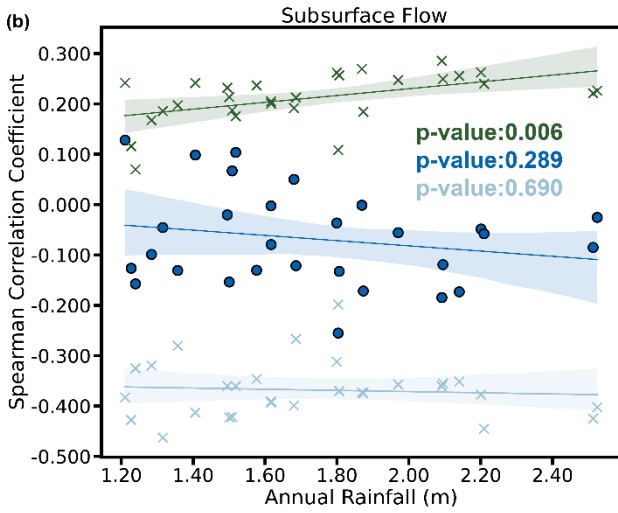
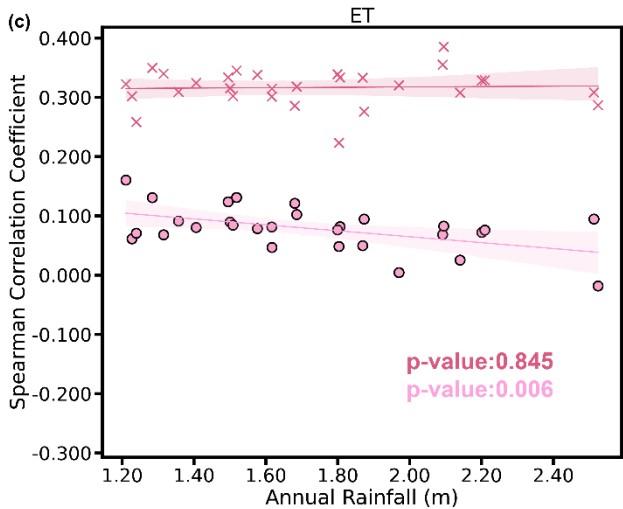
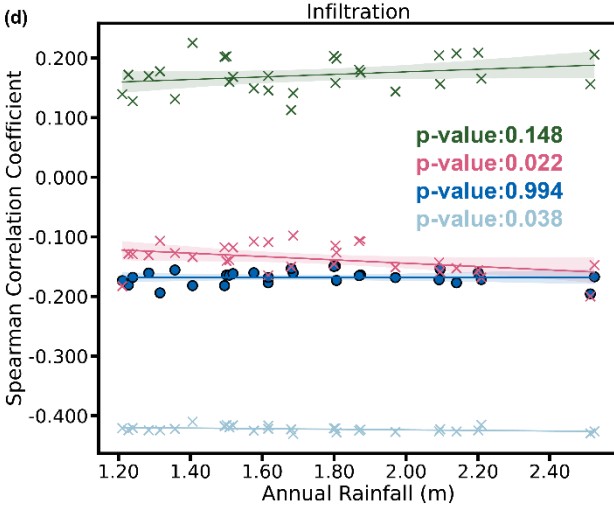

**Figure 12.** Scatter plots of Spearman statistic values of slope and four model outputs under 29 years of different annual rainfall amounts, with statistical significance levels indicated by p-values in plots. Shaded areas indicate 95% confidence intervals. Regression equations for **surface runoff** (a) NTRB: Zone3 (y=0.03x+0.08); BJRB: Zone1 (y=-0.02x-0.23), Zone3 (y=0.01x+0.3). **Subsurface flow** (b) NTRB: Zone3 (y=-0.05x+0.02); BJRB: Zone1 (y=0.07x+0.09), Zone3 (y=-0.01x-0.35). **ET** (c) NTRB: Zone2 (y=-0.05x+0.16); BJRB: Zone2 (y=0.003x+0.31). **Infiltration** (d) NTRB: Zone3 (y=0.000x-0.17); BJRB: Zone1 (y=0.02x+0.13), Zone2 (y=-0.03x-0.09), Zone3 (y=-0.005x-0.41).

## 4.3.2 Rainfall intensifies the changes in groundwater caused by LULCC

Figure 10 presents scatter plots correlating KS test values for four hydrological outputs with 29 years of annual rainfall data,

evaluating how the impacts of LULCC vary under different precipitation intensities. Our analysis highlights significant

variability in the effects of LULCC across various annual rainfall amounts in the BJRW. Here, surface runoff and infiltration exhibit reduced variations before and after urbanization as annual rainfall increases, whereas variations in subsurface flow exhibit greater magnitude with increasing annual rainfall. In the NTRW, the most obvious changes are observed in annual subsurface flow, which also shows increased variation with higher levels of annual precipitation. In scenarios where all surfaces are permeable, an increase in annual rainfall leads to progressive soil saturation, consequently enhancing surface runoff and reducing water infiltration. This pattern is similar to that observed on impervious surfaces. As annual rainfall increases, the disparities in surface runoff and infiltration between different land use patterns diminish. However, the impact on subsurface flow differs between permeable and impervious surfaces. In areas with high permeability, increased rainfall promotes soil saturation, enhancing subsurface flow. However, in areas dominated by impervious surfaces, limited infiltration capacity restricts groundwater recharge, resulting in poor saturated zone connectivity and reduced subsurface flow. These contrasting responses lead to more substantial differences in subsurface flow patterns between different land use types as annual rainfall increases.

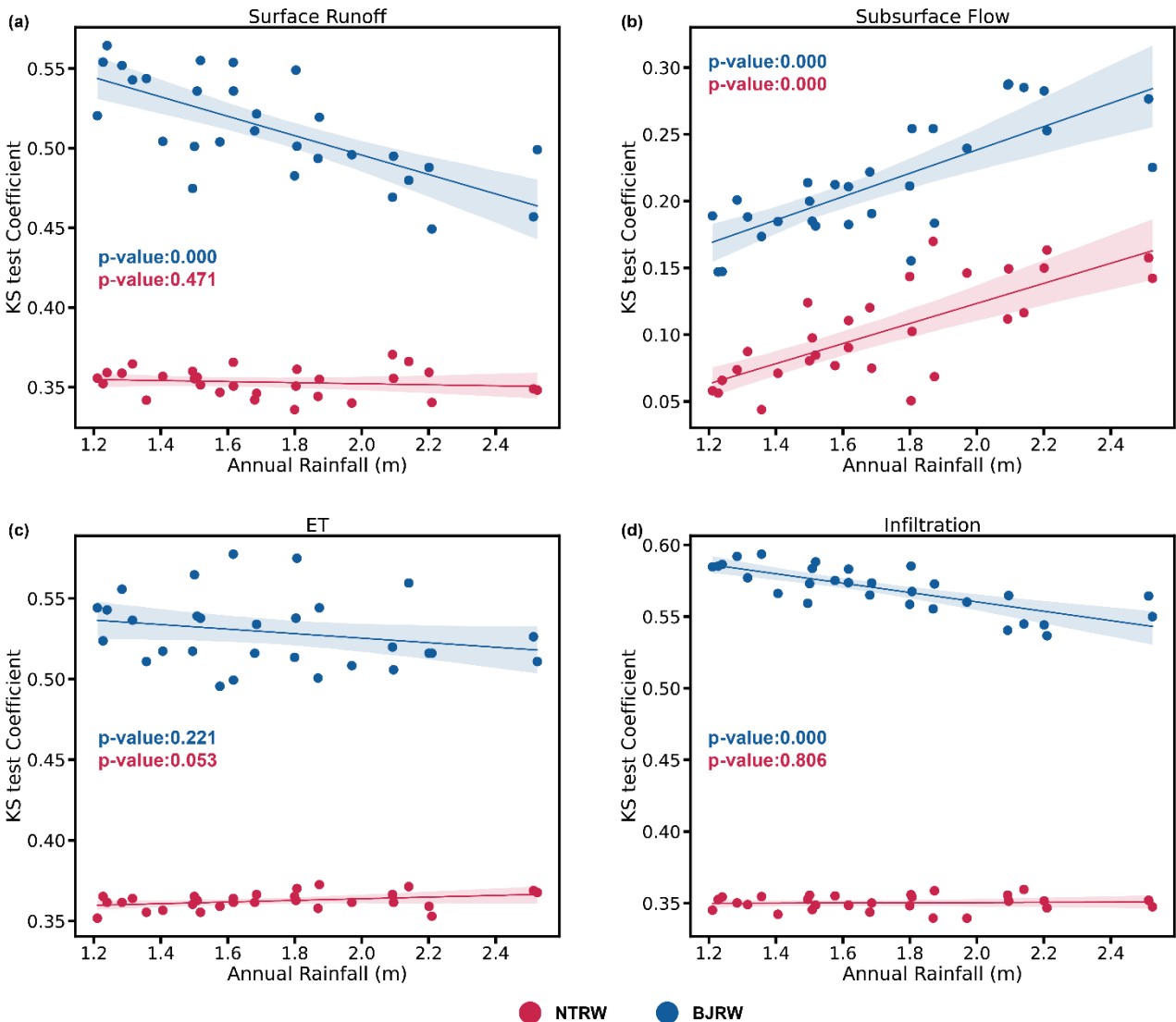

**Figure 13.** Scatter plots of KS test coefficients between LULCC and four model outputs under 29 years of different annual rainfall amounts, with statistical significance levels indicated by p-values in plots. Shaded areas indicate 95% confidence intervals. Regression equations for **surface runoff** (a) NTRW (y=0.36-0.003x); BJRW (y=0.62-0.06x). **Subsurface flow** (b) NTRW (y=-0.03+0.08x); BJRW (y=0.06+0.09x). ET (c) NTRW (y=0.35+0.01x); BJRW (y=0.55-0.01x). **Infiltration** (d) NTRW (y=0.35+0.001x); BJRW (y=0.63-0.03x).

## 4.4 Further discussion

### 4.4.1 Patterns of surface and subsurface hydrological behavior

Surface and subsurface hydrological processes exhibit distinct differences in their temporal responses and controlling factors. Surface runoff and infiltration respond rapidly and intensely to rainfall events, primarily driven by precipitation at daily timescales, making it difficult to identify stable topographic controls. However, when extending to annual timescales, these quick-response processes gradually reveal their sensitivity to slope and elevation patterns. In contrast, subsurface hydrological processes show weaker direct responses to rainfall, instead relying more heavily on topographic features and upstream water contributions to determine flow patterns.

This research further demonstrates that integrated indicators like the TWI exhibit more pronounced predictive significance for soil moisture patterns at longer (annual) timescales (Seibert et al., 2003; Rinderer et al., 2014; Kopecký et al., 2021). At this temporal scale, soil moisture and groundwater distribution reach a relatively stable state, making topographic influences on both surface and subsurface hydrological processes more evident.

Additionally, urbanization-induced expansion of impervious surfaces has significantly altered surface hydrological processes, with impacts varying across regions and topographic conditions. In contrast to surface processes, urbanization's effects on subsurface flow are less pronounced (Fig. 11), with the most significant changes occurring in low-elevation regions (Fig. 10c), consistent with the findings of Siddik et al. (2022).

**4.4.2 Suggestions for urban water resource management**

Urban hydrology is a highly complex issue (McGrane, 2016; Qi et al., 2021). This research indicates that urban hydrological processes are influenced not only by local topography but also by the characteristics of the entire watershed. The effects of urbanization are not uniform but rather distinctly localized, with varying intensities across different spatial areas.

Cities located in steep and rainy watersheds like Hong Kong face more severe challenges. Due to its steep mountainous terrain and limited flat regions, Hong Kong has minimal zones suitable for stable water storage (Chen, 2001). Additionally, with its subtropical monsoon climate bringing intense rainfall during typhoon seasons, Hong Kong faces significant urban flooding risks in its low-elevation, high-density building regions (He et al., 2021; Yang et al., 2022). Although flat cities like Shenzhen have the capability of buffering the effects of urbanization through flatter topography, their high level of urbanization still poses significant challenges for flood management under extreme precipitation conditions. Constrained by space limitations, development has extended into floodplains, wetlands, and reclaimed coastal zones (Chan et al., 2014).

Evidence suggests that depending exclusively on hard-engineering infrastructure for urban flood defense is both costly and impractical (Chan et al., 2022; Cai et al., 2021). The role of non-structural flood control measures should be emphasized, including public participation and training, the development of comprehensive water resource monitoring networks, and hydrological models for more precise flood monitoring and prediction. Technology-driven warning systems have demonstrated their effectiveness in predicting urban flood risks (Yereseme et al., 2021). The experience of sustainable flood risk management in the UK, Netherlands, USA, and Japan provides useful lessons for developed cities worldwide (Chan et al., 2022). The use of hydrological modeling to combine flood risk assessment with urban planning leads to more resilient urban water management systems. In particular, the application of ISSHMs can greatly enhance predictive capabilities before implementing land-use changes. By calibrating models to reflect current watershed conditions, planners can readily simulate various "what-if" scenarios to evaluate how proposed urban development patterns might alter hydrological processes.

**4.4.3 Limitation and future work**

Our study provides valuable insights into the effects of topography and LULCC on hydrological processes across various spatiotemporal scales in different watersheds. Although the hydrological model used was comprehensively calibrated using observational data and demonstrated accurate predictive capabilities, several limitations warrant consideration. Firstly, the calibration of the model parameters was conducted manually using local data, which may not encompass the optimal parameter sets unidentified in this study. Furthermore, the inherent uncertainties associated with the monitoring data and the model structure were not thoroughly evaluated. Due to the complexity of ISSHMs and the significant amount of time required to thoroughly assess all uncertainties, such evaluations remain challenging but are necessary for advancing the field. Secondly, our study area is located in a subtropical humid region characterized by frequent rainfall and consistently moist soils. This geographical specificity may limit the generalizability of our findings to regions with different climatic conditions. And the rainfall data utilized in this study only encompassed the typical range of precipitation for the region; extreme rainfall events, which may induce unique hydrological responses, were not investigated. The impact of such extreme conditions remains to be explored in future studies. Finally, the ET process differs from other three processes as it is influenced not only by land cover but also by climatic factors such as solar radiation, temperature, and humidity (Blyth, 1999). Our findings indicate that establishing a clear, general relationship between topography and ET is difficult. However, the analysis of LULCC and ET shows that converting forested areas into built-up land reduces the total ET at the watershed scale (Fig. 11). Since our research primarily focuses on terrestrial hydrological processes, the discussion of ET remains relatively limited.

**5 Conclusions**

Utilizing the ISSHM model, SHUD, this study explored the effects of topographical slope and urbanization-induced LULCC on surface runoff, subsurface flow, ET, and infiltration across various spatiotemporal conditions in two neighboring subtropical watersheds. Our findings reveal that both local topography (specifically local slope) and overall watershed topography significantly influence hydrological processes across different temporal and spatial scales. At the daily scale, precipitation emerges as the dominant control factor for rapid hydrological processes (infiltration and surface runoff), with local slope having limited influence. However, for slower processes like subsurface flow, local slope demonstrates a notable impact. At the annual scale, local slope correlates with both fast and slow hydrological processes in high-elevation areas. In low-elevation regions, the relationship between local slope and hydrological processes is more complex: flat watersheds show clear correlations between local slope and hydrological processes, while in steep watersheds, low-elevation hydrological processes might be more influenced by upstream contributions rather than local terrain slope.

The varying influences of local and overall watershed topography lead to spatially differentiated impacts of LULCC.

Urbanization significantly increases surface runoff while decreasing infiltration and ET, with minimal impact on subsurface

flow. Per unit of urbanized area, watersheds with gentler slopes demonstrate a greater capacity to mitigate LULCC effects,

particularly in reducing the magnitude of increased surface runoff. However, this buffering capacity diminishes as annual

precipitation increases. Additionally, the difference in subsurface flow between pre- and post-urbanization conditions becomes

more pronounced with increased annual precipitation. This study underscores the importance of incorporating non-structural

approaches in urban water management. Well-calibrated ISSHM models have demonstrated their practical value in land-use

scenario design, enabling rapid simulation of how different development patterns affect hydrological processes across temporal

and spatial scales. The integration of such hydrological modeling with urban planning will help build more resilient cities.

**Appendix A: SHUD hydrological processes formulas**

The comprehensive exposition of the governing equations for the SHUD is provided in Shu et al. (2020). Here, the emphasis

is placed on expounding the equations that are relevant to the processes addressed in this study.

- *Infiltration.* SHUD adopts the Richards equation like most ISSHMs adopted to describe the infiltration process. While there

are no general analytical solutions to the Richards equation, SHUD adopted the Green-Ampt infiltration equation (Eq. (A1)),

which allows a simple form of Darcy's law to be used to calculate the infiltration rate $q_i$ [LT$^{-1}$],

$$q_{\mathrm{i}} = K_{\mathrm{i}} \left( 1 + \frac{h_{\mathrm{s}}}{D_{\mathrm{inf}}} \right)$$    (A1)

where $h_{\mathrm{s}}$ [LT$^{-1}$] is the ponding water height plus precipitation, $D_{\mathrm{inf}}$ [L] is the infiltration depth representing the top soil layer,

$K_i$ [LT$^{-1}$] is the effective infiltration conductivity, and it is a function of soil saturation ratio, soil properties, and $h_{\mathrm{s}}$. The Green-

Ampt method assumes that the infiltrating wetting front forms a sharp jump from a constant initial moisture content ahead of

the front to saturation at the front.

-*Evapotranspiration.* Potential evapotranspiration (PET) is computed using the Penman-Monteith equation (Eq. (A2)), while

actual evapotranspiration (AET) is derived by multiplying PET with a soil moisture stress coefficient, determined by soil

moisture content and groundwater table depth.

$$\lambda E = \frac{\Delta_e H + \rho_a c_p (e_s(T_z) - e_z)/r_a}{\Delta_e + \gamma(1 + r_c/r_a)},$$    (A2)

where $\lambda (= 2.47 \times 10^6$, Jkg$^{-1}$) is the latent heat of evaporation, $E$ [LT$^{-1}$] is the PET rate, $\Delta_e$ is the slope of the saturation vapor

pressure versus temperature curve, $H$ is total available energy, $\rho_a$ is the density of the air, $c_p$ is the specific heat capacity of the

air, $e_s(T_z)$ is the saturated vapor pressure at the height of $z$, $e_z$ is the vapor pressure at the height of $z$, $r_a$ and $r_c$ are the two

resistance coefficients, $\gamma$ is the psychrometric constant.

- *Surface runoff.* The kinematic wave equation (Eq. (A3)) is used to approximate the surface runoff in the SHUD,

$$\frac{\partial h}{\partial t} = -\frac{\partial (vh)}{\partial x} - \frac{\partial (vh)}{\partial y} + r, \tag{A3}$$

where $h$ [L] represents the average depth of flow, $v$ [LT$^{-1}$] is the flow velocity, and $r$ [LT$^{-1}$] is a rate of addition or loss of water

caused by precipitation, infiltration and evaporation. The relationship between $v$ and $h$ is represented by the Manning equation

(Eq. (A4)),

$$v = -\frac{S_0^{\frac{1}{2}} h^{\frac{3}{5}}}{n}, \tag{A4}$$

where $S_0$ [-] is the surface slope, $n$ [TL$^{-1/3}$] is the Manning roughness.

- *Subsurface flows.* The SHUD applies the Richards equation (Eq. (A5)) to describe both saturated and unsaturated flows, and

the water density is assumed to be constant,

$$\frac{\partial \theta}{\partial t} = \frac{\partial}{\partial x}\left[K_x(\theta)\frac{\partial \Phi}{\partial x}\right] + \frac{\partial}{\partial y}\left[K_y(\theta)\frac{\partial \Phi}{\partial y}\right] + \frac{\partial}{\partial z}\left[K_z(\theta)\frac{\partial \Phi}{\partial z}\right], \tag{A5}$$

where $\theta$ [-] is volumetric moisture content, $K_x(\theta)$ [LT$^{-1}$], $K_y(\theta)$ [LT$^{-1}$], and $K_z(\theta)$ [LT$^{-1}$] indicate hydraulic conductivity

depends on direction and is treated as a function of $\theta$, $\Phi$ [L] is the total potential ($\Phi = \psi + z$ where $\psi$ [L] is the capillary

potential and $z$ is the elevation above the datum). The SHUD utilizes the van Genuchten functions to solve the relationship for

soil moisture content, capillary potential, and hydraulic conductivity.

**Appendix B: Assessment equations**

The Spearman's rank correlation method evaluates the strength and monotonic nature of relationships between two variables

without relying on assumptions regarding data distribution or residuals. The KS two-sample test compares two samples to

determine if they are drawn from the same distribution, without assumptions about the underlying distribution. The KS statistic

is the maximum absolute difference between the CDFs of the two data vectors.

For the daily scale analysis, we focused on positive model outputs during rainy days (precipitation $\geq 0.1$ mm per day).

We employed matrix **D** for each zone (Zone 1 to Zone 3) to assess daily outputs related to slope angle for each grid (Eqs. (B1)

and (B2)). Model outputs for surface runoff and subsurface flow (m³ d$^{-1}$) represent net flow amounts per mesh grid. For daily

analysis, these outputs were summed to total flow volumes (m³) and divided by grid area to obtain flow depths (m). Infiltration

and ET outputs (m d$^{-1}$) were similarly summed to daily depths (m). These standardized depths were used to analyze impacts

of slope and LULCC.

$$\mathbf{D} = \begin{bmatrix} \mathbf{W}_1 \\ \vdots \\ \mathbf{W}_N \end{bmatrix}, \tag{B1}$$

$$\mathbf{W}_n = \begin{bmatrix} y_{1n} & p_1 & s_n \\ \vdots & \vdots & \vdots \\ y_{in} & p_i & s_n \end{bmatrix} \tag{B2}$$

In matrix $\mathbf{D}$, each row $\mathbf{W}_n$ ($n$=1, 2, …, $N$) corresponds to the model outputs associated with a specific hydrological process

of the $n$th grid. Within $\mathbf{W}_n$, each row represents a rainy day under consideration, with $i$ denoting the total number of rainy days

analyzed. Each row comprises three values: the daily model output $y_{kn}$ ($k$=1, 2, …, $i$), the corresponding rainfall amount $p_k$

($k$=1, 2, …, $i$), and the grid's slope angle $s_n$. Consequently, the Spearman correlation coefficient was computed between the

transpose vectors $\boldsymbol{y}^T_{N \times i}$ and $\boldsymbol{s}^T_{N \times i}$.

To analyze LULCC effects, vectors $\boldsymbol{H}_d$ (Eq. (B3)) and $\boldsymbol{C}_d$ (Eq.(B4)) were generated for each grid under HLU and CLU

patterns, and the KS test value was computed between these two vectors for each grid,

$$\boldsymbol{H_d} = [y_{1|\text{HLU}}, y_{2|\text{HLU}}, \cdots, y_{i|\text{HLU}}], \tag{B3}$$

$$\boldsymbol{C_d} = [y_{1|\text{CLU}}, y_{2|\text{CLU}}, \cdots, y_{i|\text{CLU}}], \tag{B4}$$

where $i$ denotes the total number of rainy days, $y_{k|\text{HLU}}$ and $y_{k|\text{CLU}}$ ($k$=1, 2, …, $i$) represent the model daily output of this grid

under the HLU pattern and the CLU pattern on the $k$th rainy day, respectively. We also calculated the absolute difference in

mean values of these two vectors to quantify the magnitude of change between the two land use patterns in terms of their

effects on the model outputs.

To evaluate the effects of slope on an annual scale, a new matrix $\mathbf{A}$ is constructed as following Eq. (B5):

$$\mathbf{A} = \begin{bmatrix} y_1 & s_1 \\ \vdots & \vdots \\ y_N & s_N \end{bmatrix}, \tag{B5}$$

where $N$ represents the number of grids within each zone, $y_n$ and $s_n$ (n=1, 2, …, $N$) denote the annual output and slope angle

for the $n$th grid, respectively. Only grids with annual volumes exceeding 10 mm were considered for surface runoff and

subsurface flow analysis to concentrate on pronounced flows. Subsequently, the Spearman correlation coefficient was

calculated between the transpose vectors $\boldsymbol{y}^T_N$ and $\boldsymbol{s}^T_N$.

The KS test was also applied at the annual scale to compare model outputs between HLU and CLU patterns across the

entire subbasin range. Vectors $\boldsymbol{H}_y$ (Eq. (B6)) and $\boldsymbol{C}_y$ (Eq. (B7)) represent the annual model outputs under the HLU pattern and

the CLU pattern, respectively.

$$\boldsymbol{H}_y = [y_{1|\text{HLU}}, y_{2|\text{HLU}}, \cdots, y_{j|\text{HLU}}], \tag{B6}$$

$$570 \quad \boldsymbol{C}_y = [y_{1|\text{CLU}}, y_{2|\text{CLU}}, \cdots, y_{j|\text{CLU}}], \tag{B7}$$

Here $z$ denotes the number of grids across each subbasin, $y_{k|\text{HLU}}$ and $y_{k|\text{CLU}}$ ($k$=1, 2, …, $z$) represent the model annual

output of the $k$th grid under the HLU pattern and the CLU pattern, respectively. Then the KS test was carried out between these

two vectors.

**Appendix C: Supplementary calibration results**

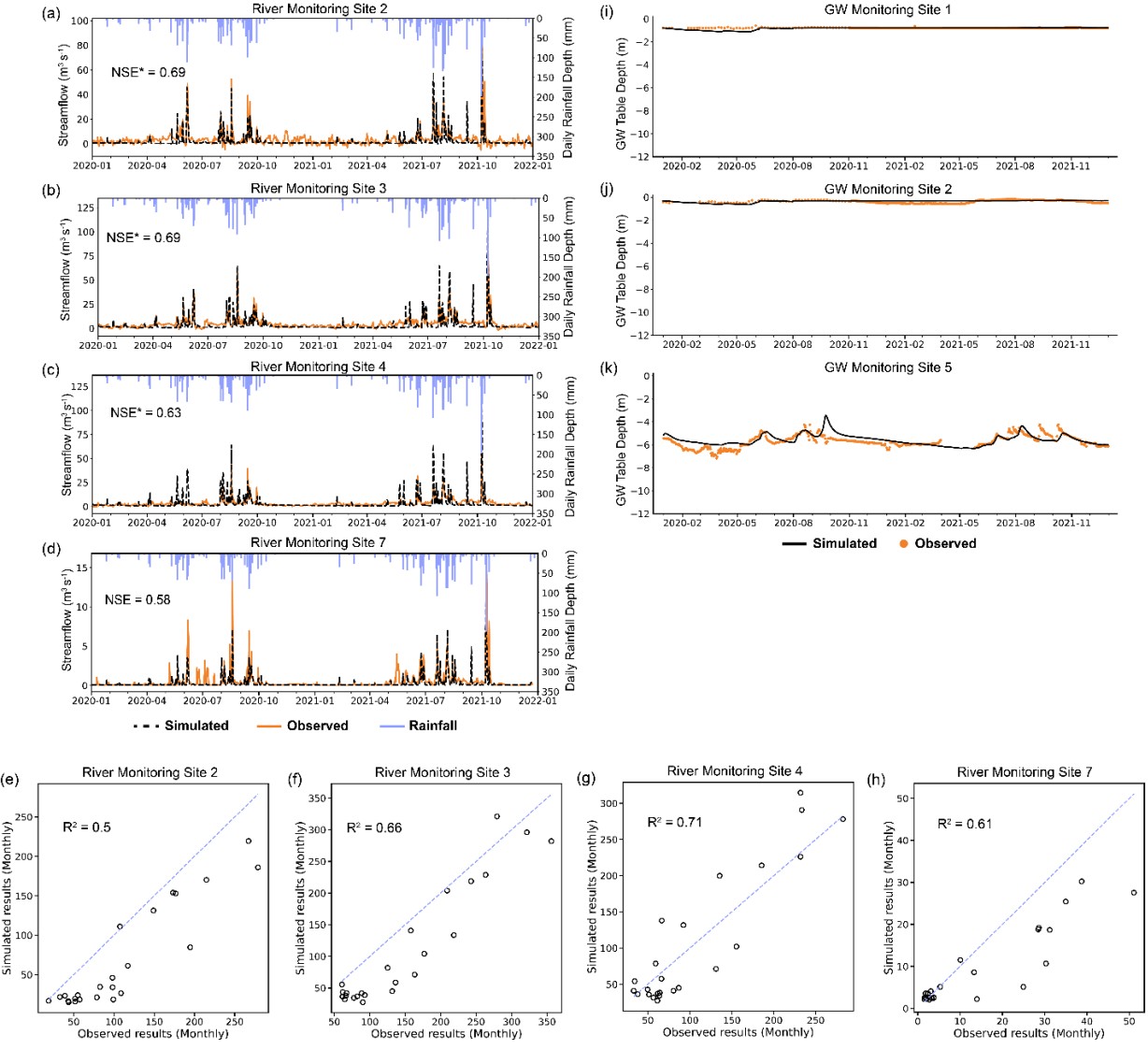

**Figure C1.** Other sites calibration results across daily river discharge (a)–(d), and monthly river discharge (e)–(h), and groundwater table depth (i)–(k).

*Code and data availability.* The source code of the SHUD model can be downloaded from https://github.com/SHUD-System/SHUD. The model spatial input data are freely available from the described source listed in Table 1. The

meteorological data and monitoring data in this study can be obtained upon request. Other related data supporting this study have been uploaded to the Zenodo repository and are accessible via the provided DOI link (10.5281/zenodo.14539888).

*Author contributions.* HL contributed to methodology, validation, visualization, writing of the original draft and editing. HY contributed to data collection, reviewing and editing the original draft. MG contributed to conceptualization, supervision, methodology, writing, reviewing and editing the original draft.

*Competing interests.* The authors declare that they have no conflicts of interest.

*Acknowledgements.* This work is financially supported by General Research Fund projects (No. 17210923).

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
