# Peer review of "hydrological signatures: a comparative study of two adjacent"

_Hydrology and Earth System Sciences, 2024_

## Referee Comment (RC2)

This article is a comparative analysis of two adjacent watersheds using the integrated surface-subsurface hydrological model (ISSHM) to examine the effects of topography and land use/land cover change (LULCC) on hydrological processes within the Greater Bay Area (GBA) of China. In general, the datasets used and methodology for the analyses are clear and appropriate. The results are well-organized. In the conclusion part, by mentioning the importance for hydrologic management strategies to consider the specific topography and LULCC characteristics of each watershed, the high-level significance of the study stands out.

However, if the significance of the study could be fully investigated with more in-depth thoughts about the results and conclusion, the publication would represent a more substantial contribution to scientific progress in the field.

The four scenarios using HLU and CLU seems intuitive and simple to apply, which might become a lightly novel or highlight of the methodology that could be potentially widely-applied. However, insufficient details about "all built-type land uses" and the process of "The HLU pattern involves reverting all built-type land uses in both watersheds to their pre-construction conditions" are provided. This is not addressed explicitly in the discussion or supplementary, which could become a valuable contribution to future studies.

Moderate to major revision is suggested to this manuscript.

Specific comments are listed below:

1) Introduction:
   a. L73-74: consider adding reference to support the statement "they are mainly based on single and spatially homogeneous watersheds".
   b. L77: may consider adding one or two sentences to describe "crucial economic zone" in China to highlight the importance of the study area.

2) Methodology:
   a. In section 3.4, the four scenarios using HLU and CLU seems intuitive and straightforward to apply, which might become a highlight of the methodology that could be potentially widely-applied. However, insufficient details about "all built-type land uses" and the process of "The HLU pattern involves reverting all built-type land uses in both watersheds to their pre-construction conditions" are provided. This is not addressed explicitly in the discussion or conclusion, and could be a valuable contribution in future studies.
   b. L187: consider adding reference of 40 meter, at which Zone 1 and Zone 2 are divided.
   c. L192-203: consider having a flowchart or a table instead of a long paragraph to demonstrate the key factors/processes, could also help the readers better understand the results.

3) Results:
   a. Figure 8: the text in the x and y-axis are too small to read.
   b. Figure 9: the current figure is busy with many equations/texts embedded with the dots, may considering make it concise by leaving the p-values in the figure but putting the equations in the captions or texts.
   c. Figure 10: similar to Figure 9

4) Conclusion:

L373-379: the author may consider adding more specific examples or related refences about how hydrologic management or local watershed agencies could use this study to improve their methodology and strategies. Thus, the application of this publication could not only benefit not only the future studies in academia but also shed light on the practical water management or engineering world.

5) Discussion (e.g., the Limitation and future work section in the current manuscript) can be substantially improved.
   a. It lacks the in-depth analysis of evapotranspiration (ET), which correlates with climatic factors such as solar radiation, temperature, humidity, etc. It is worth noting that ET is different from surface runoff, subsurface flow, and infiltration. The author may consider adding sentences in the limitation/discussion section.
   b. Groundwater dynamics was mentioned in other sections except the discussion part, considering address it in section 4.4.
   c. L258-262: consider adding text about the statement that "topographic indices more accurately reflect hydrological responses under steady-state conditions" in discussion part.
   d. Considering the study areas are the important economic zone, the author may consider relating it to other economic zone or highly-urbanized areas in other regions of the world. It may worth thinking and adding texts about how this publication could shed light on the practical water management or engineering world.

6) The abstract is also suggested to revise based on the updated revision.

---

## Author Comment (AC1)

Dear Prof. Shu,

Thank you for your thorough and insightful review of our paper. We appreciate your positive feedback on our experimental design, credibility of arguments, clarity of charts, and readability of our writing. We also acknowledge your constructive comments and suggestions for improving our study. To distinguish your comments from our responses, we will present your comments in blue bold font and our responses in regular black font. Please find our detailed responses to your concerns below:

1. **The authors explored the relationship between slope and several hydrological processes. I did not see the process of slope calculation; therefore, I am uncertain if the slope here refers to the slope calculated from the DEM in different zones or the slope calculated based on the triangular mesh of the SHUD model. Since the resolution of the triangles is lower than that of the DEM, I believe that the slope based on triangles better reflects the impact of slope on hydrological processes. According to lines 187-189, the three zones are divided based on elevation differences, but the average slope attribute within the three zones is not displayed or discussed. Therefore, the article needs to clarify the slope calculation method.**

   We appreciate your observation regarding the slope calculation. In our study, the slope was indeed calculated based on the triangular mesh of the SHUD model rather than the three zones. Specifically, we used the DEM raster file with a spatial resolution of 30m x 30m to determine the slope of each raster grid. We then calculated the average slope for each triangular mesh by averaging the slopes of all raster grids within the mesh. All subsequent analyses were conducted using the slope values of each triangular mesh, as detailed in Appendix B. We agree that this method, with its higher resolution, more accurately reflects the impact of slope on hydrological processes compared to the three zones. We will include a detailed explanation of the slope calculation method in the revised manuscript. Additionally, we will discuss the average slope attributes within the three zones based on elevation differences.

2. **Based on the SHUD model, the authors analyzed the impact of slope on hydrological processes (subsurface). Local subsurface flow is usually affected by soil characteristics, especially hydraulic conductivity, and is also influenced by slope runoff (slope + accumulating area). Therefore, the groundwater level is higher in the flat areas around the river channel, leading to a larger subsurface flow.**

   We agree with your assessment that local subsurface flow is influenced by soil characteristics and slope runoff (slope + accumulating area). Consequently, groundwater levels are higher in flatter areas, leading to larger subsurface flow. Our results support this conclusion. The daily scale subsurface flow result (Fig. 5) shows a positive correlation between subsurface flow and slope in the flatter Zone 1, whereas this correlation is less pronounced in the steeper Zones 2 and 3. This indicates that in the flatter Zone 1, as the slope increases, the subsurface flow (outflow) also increases. Additionally, the annual scale subsurface flow results (Fig. 6) demonstrate that the positive correlation between subsurface flow and slope is more pronounced in the overall flatter BJRB compared with the steeper NTRB. We will add more detailed discussions on these findings in the revised paper and include additional figures if necessary to further illustrate the relationship between subsurface flow and slope.

3. **The SHUD model outputs subsurface flow, which may include the flow (Q1, Q2, Q3) in three directions of the triangle (positive outward) and the sum/net flow in the three directions (Qs = Q1 + Q2 + Q3, or eleqsub = eleqsub1 + eleqsub2 + eleqsub3). Then, the water balance of the saturated zone of this unit should be dS = Q1 + Q2 + Q3. In the long-term trend, dS should be close to zero. Here, dS is equivalent to Qs (eleqsub). Therefore, subsurface flow should be the sum of positive flows or the sum of negative flows. If the authors**

**directly use new-flow (Qs/eleqsub) in the calculation, it no longer conforms to the meaning of subsurface flow. For example, the flow rates of the three sides of a triangle are eleqsub1 = 200, eleqsub2 = 100, and eleqsub3 = -310, then eleqsub = 200 + 100 + (-310) = -10. It indicates that the unit gains 10 units of water. In this case, the flow rate at this unit should be 300 (200 + 100) or -310, but definitely not -10. Similarly, the calculation of surface runoff may have the same concerns. I suggest that the authors clarify the reading and processing of variables in the appendix or supplementary materials.**

Firstly, we would like to thank the reviewer for the very clear and detailed explanation. We acknowledge the importance of accurately representing subsurface flow calculations to ensure our results are clear and understandable for readers. We will include a detailed explanation of the subsurface flow calculation method in the supplementary materials.

We recognize the concern regarding the use of the net flow value (e.g., -10) instead of positive or negative direction flows (e.g., 300 or -310). In this research, we use the net flow (eleqsub) to represent the water balance within the control volume. While this approach simplifies the explanation of subsurface flow dynamics, it aligns with our study's objective to examine broader hydrological trends and patterns rather than the detailed hydraulic dynamics within each mesh. As mentioned in Appendix B, page 21, line 418 of the manuscript, we only calculated the triangular meshes with a positive net flow, indicating subsurface or surface net outflow. This ensures our analysis focuses on areas contributing to water movement out of the system, consistent with our objective to better understand outflow behavior.

To address your suggestion, we will provide a detailed explanation of the variable reading and processing methods in the appendix or supplementary materials. This will clarify the rationale behind using net flow and discuss its implications.

4. **Parameter calibration is a very challenging task for any ISSHMs. Although your study uses manual parameter calibration, I strongly recommend that you share more calibration details/experiences so that others can learn from your work. For example, how were sensitive parameters determined? How were parameters adjusted to gradually approach the observed results? How was it determined that the current parameter set reached "optimal" or "usable" levels?**

We appreciate the suggestion to provide more details about the parameter calibration process.

First, regarding the identification of sensitive parameters, we combined our previous experience with ISSHM model calibration and referenced relevant literature to select the seven parameters listed in Table 3. For example, Liu et al. (2020) considered the impact of climate variations, aquifer thickness, unsaturated zone parameters (such as soil saturated hydraulic conductivity and VG parameters), aquifer parameters (unconfined and confined aquifer hydraulic conductivity), and surface water runoff parameters (the length of the flow path for runoff contribution to the overland flow domain) on stream baseflow. Their findings indicated that the unsaturated zone parameters and aquifer parameters had the greatest influence on ET and baseflow. In this study, we selected parameters from the unsaturated zone and aquifer categories. Additionally, we observed the results of changing individual parameters within the unsaturated zone or aquifer (setting a series of variations) to assess their impact on streamflow and baseflow. We found that changes in certain parameters had minimal impact, leading us to exclude the remaining parameters in these two categories. Ultimately, we identified the seven most sensitive parameters for further calibration.

The iterative adjustment process to align the simulation results with the observed results can be roughly divided into three stages:

- Coarse adjustment of streamflow: During this stage, we primarily referenced the monitoring results of streamflow. The goal was to align the trend, peak timing, and peak values of the simulation results with the monitoring results, even if the baseflow simulation results were not yet consistent with the observations.

- Fine adjustment of streamflow: In this stage, we focused on adjusting parameters related to the aquifer to ensure that the simulated baseflow process closely matched the monitoring results.

- Adjustment of groundwater tables: In the final stage, we aim to minimize extensive modifications to previously determined parameters, instead focusing on adjusting soil and aquifer parameters near the monitoring points.

Throughout all stages of iterative parameter adjustment, we referred to the empirical ranges of the corresponding soil or aquifer values and their relative magnitudes to ensure the reasonableness of the parameter values.

Regarding the criteria for determining the "optimal" or "usable" parameter set, after completing the third stage of calibration, we calculated the NSE index for the simulated and observed streamflow results. If NSE > 0.5 and the simulated groundwater tables were also within an acceptable range (assessed using expert judgment), we considered this parameter set usable. Otherwise, we repeated the second and third stages. By consistently considering the reasonable ranges of parameters and the relative magnitudes between different soil and aquifer layers throughout the iterative adjustment process, we ensured that the parameters meeting our criteria were all usable.

In the revised manuscript, we will include a comprehensive description of our manual parameter calibration approach.

5. **Although the authors have clarified the data sources of this study, I strongly recommend that the authors share the research data of this article (including model input files, model source code used to implement this study, and calibrated parameters) in an open database (such as Zenodo). Of course, you can desensitize or not share meteorological and hydrological data that are restricted by copyright or confidentiality clauses. Sharing this data can help other researchers replicate your work and further expand the impact of your work. This is just a suggestion.**

Thank you for your thoughtful suggestion regarding data sharing. We are happy to share the model input files, source codes, and calibrated parameters in open databases such as Zenodo. We believe that sharing these resources will help other researchers replicate our work and further expand the impact of our study.

**References**

Liu, H., Dai, H., Niu, J., Hu, B. X., Gui, D., Qiu, H., Ye, M., Chen, X., Wu, C., Zhang, J., and Riley, W.: Hierarchical sensitivity analysis for a large-scale process-based hydrological model applied to an Amazonian watershed, Hydrol. Earth Syst. Sci.,24, 4971–4996, https://doi.org/10.5194/hess-24-4971-2020, 2020.

---

## Author Comment (AC2)

Dear Reviewer,

Thank you for your comprehensive and detailed evaluation of our manuscript. Your valuable comments on the importance of this research are of great significance to us. And we will do our best to improve the depth of the analysis and make this research a more significant contribution to the field.

To distinguish your comments from our responses, we will present your comments in blue bold font and our responses in regular black font. Please find our detailed responses to your concerns below:

1. **The four scenarios using HLU and CLU seems intuitive and simple to apply, which might become a lightly novel or highlight of the methodology that could be potentially widely-applied. However, insufficient details about "all built-type land uses" and the process of "The HLU pattern involves reverting all built-type land uses in both watersheds to their pre-construction conditions" are provided. This is not addressed explicitly in the discussion or supplementary, which could become a valuable contribution to future studies.**

Thank you for your valuable feedback. We understand the importance of providing detailed information on the treatment of land use types and the reversion of built-up land to its pre-construction state.

As for the **current land use pattern** construction, we utilized the 2020 10-meter resolution raster data from the Dynamic World Project (Fig. R1a). For each triangular mesh grid in our study area, we assigned the land use type that occupies the largest proportion of the area within that grid. This approach ensures that each mesh grid is classified based on its dominant land use, resulting in the current land use pattern (Fig. R1b).

To derive the **historical land use pattern**, we modified the current land use data by converting all mesh grids classified as built-type land uses within both watersheds to the trees land use type (Fig. R1c). This transformation effectively reverts built areas to their pre-construction conditions, allowing us to simulate a scenario without urban development.

Both the original raster data and our model incorporate eight distinct land use types, as outlined in the legend of Fig. R1. Each land use type is assigned specific parameter values within the model, including leaf area index (LAI), albedo, roughness, root zone depth, and impervious area fraction. For instance, for the type of trees land use, the impervious area is 0%, while for the type of built land use, the impervious area is 94%. We acknowledge that the initial manuscript lacked sufficient detail regarding the treatment of built-type land uses and the methodology for reverting them to historical conditions. In the revised version, we will describe the method more comprehensively and explain the parameters in the model.

[Figure]

**Figure. R1 Land use types within the study area, with the base map provided by the 2020 10-meter resolution raster data from the Dynamic World Project (a). Current land use pattern in the model (b). Historical land use pattern in**

**the model (c).**

2. **L73-74: consider adding reference to support the statement "they are mainly based on single and spatially homogeneous watersheds".**

Thank you for your valuable suggestion. We will make an effort to add appropriate citations to support this claim. When suitable references are limited, however, we will revise the wording to more accurately reflect the current state of research and add relevant literature to ensure the accuracy and persuasiveness of the statement.

3. **L77: may consider adding one or two sentences to describe "crucial economic zone" in China to highlight the importance of the study area.**

Thank you for your valuable suggestion. We will add a description of the Greater Bay Area (GBA) to emphasize its significance as a critical economic zone. The revised sentence will be as follows: *"We simulate the hydrological processes of the two watersheds in the Greater Bay Area (GBA), a critical economic zone in China that encompasses major cities such as Guangzhou, Shenzhen, Hong Kong, and Macao. According to the latest data, the GDP of the GBA exceeded 14 trillion yuan in 2023. However, the region faces significant challenges in achieving sustainable growth and coordinated spatial planning under rapid urbanization. These factors make the GBA an ideal area for studying hydrological processes. For this study, we use the Simulator for Hydrologic Unstructured Domains (SHUD) as an ISSHM."*

4. **In section 3.4, the four scenarios using HLU and CLU seems intuitive and straightforward to apply, which might become a highlight of the methodology that could be potentially widely-applied. However, insufficient details about "all built-type land uses" and the process of "The HLU pattern involves reverting all built-type land uses in both watersheds to their pre-construction conditions" are provided. This is not addressed explicitly in the discussion or conclusion, and could be a valuable contribution in future studies.**

Thank you for your valuable comments. In the first question in the response, we have provided a preliminary explanation of "all built-up land uses" and the process of reverting them to their pre-construction state. Considering that the main focus of the study is land use change and terrain slope, we plan to restructure section 3.4 in future revisions. This section will provide more precise and detailed explanations of the four scenario settings.

5. **L187: consider adding reference of 40 meter, at which Zone 1 and Zone 2 are divided.**

Thank you for your valuable suggestion. The division of each zone was determined based on two main criteria. First, we calculated the average elevation of each triangular mesh grid using DEM raster data. Then, we applied the natural breaks method to classify all grids in the two watersheds into six elevation groups, with the natural breakpoints for the first two groups being approximately 40 meters and 120 meters. We also considered the number of mesh grids within each zone, ensuring that each zone contains enough mesh grids for reliable statistical analysis. Since the number of grids in groups three through six was too small for meaningful analysis, we combined them into a single Zone 3. We will clarify this process in the revised manuscript to ensure that this division is clearly explained.

6. **L192-203: consider having a flowchart or a table instead of a long paragraph to demonstrate the key factors/processes, could also help the readers better understand the results.**

Thank you for your helpful suggestion. We will incorporate a flowchart in the revised version of the manuscript.

7. **Figure 8: the text in the x and y-axis are too small to read**

Thank you for pointing this out. We will revise Figure 8 to enlarge the text on both the x and y-axes.

8. **Figure 9: the current figure is busy with many equations/texts embedded with the dots, may considering make it concise by leaving the p-values in the figure but putting the equations in the captions or texts.**

Thank you for your valuable suggestion. We will revise the figure by keeping the p-values within the figure and moving the equations to the caption or text to make the figure less busy and more concise.

9. **Figure 10: similar to Figure 9.**

Thank you for your suggestion. Like Figure 9, we will revise Figure 10 by keeping the p-values in the figure and moving the equations to the caption or text.

10. **L373-379: the author may consider adding more specific examples or related references about how hydrologic management or local watershed agencies could use this study to improve their methodology and strategies. Thus, the application of this publication could not only benefit not only the future studies in academia but also shed light on the practical water management or engineering world.**

Thank you for your insightful suggestion. Providing more specific examples and references would significantly enhance the usefulness of our research. In the revised manuscript, we will give some specific examples and add relevant references to illustrate the impact of these research results on practical water resources management and engineering solutions.

11. **It lacks the in-depth analysis of evapotranspiration (ET), which correlates with climatic factors such as solar radiation, temperature, humidity, etc. It is worth noting that ET is different from surface runoff, subsurface flow, and infiltration. The author may consider adding sentences in the limitation/discussion section.**

Thank you for your valuable feedback. We admit that the ET process is quite different from surface runoff, subsurface flow and infiltration processes. In addition to land factors, ET is more closely related to climatic factors such as solar radiation, temperature and humidity. Our research focuses mainly on terrestrial hydrological processes, so the discussion of ET is relatively limited. In the revised version, we will appropriately expand the discussion of ET while ensuring the accuracy of the analysis. In addition, we will address this limitation in the "limitations/discussion" section.

12. **Groundwater dynamics was mentioned in other sections except the discussion part, considering address it in section 4.4.**

Thank you for your valuable suggestion. We acknowledge the importance of groundwater dynamics in our study. The revised manuscript will address groundwater dynamics in limitations and future work section. This addition will include an analysis of groundwater processes within the different watersheds and their implications for hydrological processes.

13. **L258-262: consider adding text about the statement that "topographic indices more accurately reflect hydrological responses under steady-state conditions" in discussion part.**

Thank you for your insightful suggestion. We agree that elaborating on how "topographic indices more accurately reflect hydrological responses under steady-state conditions" will strengthen our discussion.

For example, research conducted in a Swedish till catchment demonstrated that groundwater levels near streams closely follow runoff dynamics, but this correlation diminishes with distance from the stream.

Groundwater levels in upslope areas can rise independently of streamflow, indicating a deviation from the steady-state assumption (Seibert et al., 2003). This finding highlights that groundwater flow exhibits high spatial variability over short periods. Consequently, when the model simulation period or event scale is short, it is crucial to account for the non-steady-state differences caused by recharge on groundwater flow. In the revised manuscript, we will provide a clearer and more detailed explanation.

14. **Considering the study areas are the important economic zone, the author may consider relating it to other economic zone or highly-urbanized areas in other regions of the world. It may worth thinking and adding texts about how this publication could shed light on the practical water management or engineering world**

Thank you for your valuable suggestions. Due to differences in topography and planning strategies, there are significant differences in the urban planning patterns of the two watersheds we studied. Despite these differences, Shenzhen and Hong Kong are highly economically developed cities. In the revised manuscript, we will appropriately add a sentence or two to discuss how these contrasting urbanization models and topographic features provide meaningful insights for other highly urbanized and economically developed regions worldwide.

15. **The abstract is also suggested to revise based on the updated revision.**

Yes! We will revise the abstract according to the updated manuscript.

**References**

Seibert, J., Bishop, K., Rodhe, A., and McDonnell, J. J.: Groundwater dynamics along a hillslope: A test of the steady state hypothesis, Water Resour. Res., 39, 1, https://doi.org/10.1029/2002WR001404, 2003.

---

## Author Response (AR1)

Dear editor and reviewers,

We would like to thank you all for the constructive and valuable comments, which will considerably improve the manuscript. In the new version, the modified parts are highlighted in red. Here, we provide responses to each comment by the reviewers. To distinguish reviewers' comments from our responses, we will present reviewers' comments in blue bold font, our responses in regular black font, and the revised parts in the manuscript are shown in red font.

**Referee #1**

**General comments:**

**This paper explores the impacts of land use change and basin topography on hydrological processes using the integrated surface-subsurface hydrological model (ISSHM), the SHUD model. The study conducted model simulations and analyses within the Shenzhen-Hong Kong basin. The experimental design is comprehensive, the arguments are credible, the charts are clear, and the writing is readable. However, I have some concerns that could make this study even more comprehensive and reliable.**

**REPLY:** Thank you for your thorough and insightful review of our paper. We appreciate your positive feedback on our experimental design, credibility of arguments, clarity of charts, and readability of our writing. We also acknowledge your constructive comments and suggestions for improving our study.

**Specific comments:**

1. **The authors explored the relationship between slope and several hydrological processes. I did not see the process of slope calculation; therefore, I am uncertain if the slope here refers to the slope calculated from the DEM in different zones or the slope calculated based on the triangular mesh of the SHUD model. Since the resolution of the triangles is lower than that of the DEM, I believe that the slope based on triangles better reflects the impact of slope on hydrological processes. According to lines 187-189, the three zones are divided based on elevation differences, but the average slope attribute within the three zones is not displayed or discussed. Therefore, the article needs to clarify the slope calculation method.**

   **REPLY:** We appreciate your observation regarding the slope calculation. In our study, the slope was indeed calculated based on the triangular mesh of the SHUD model instead of the original 30-meter resolution DEM data. Specifically, we extracted elevation values for each triangular mesh vertex from the 30-meter resolution DEM, re-interpolated these values to create a new raster DEM, and then calculated the average slope for each mesh grid. All subsequent analyses were conducted using the slope values of each triangular mesh, as detailed in Appendix B. We agree that this method, based on triangles better reflects the impact of slope on hydrological processes. We have included a detailed explanation of the slope calculation method in the revised manuscript. Additionally, we have separated Fig. 3 into two figures, and in the latter figure, we analyzed the slope attributes within three zones defined by elevation differences.

   Lines 225-229 (Section 3.4.2):

   "To isolate the impact of slope from LULCC effects, we analyzed slope impacts within the two watersheds exclusively under the HLU pattern. To ensure a coherent assessment of how slope influences hydrological processes, we derived slope values based on the topographical characteristics of the model instead of the original 30-meter resolution DEM data. We extracted elevation values for each triangular mesh vertex from the original 30-meter DEM data, re-interpolated these values to create a new raster DEM, and then calculated the average

slope for each mesh grid."

Lines 238-239 (Section 3.4.2):

"After classification, the mean slope values for each zone are shown in Fig. 6. Since the NTRW terrain is generally steeper, the average slope value for each zone is greater in NTRW than in BJRW."

Figure 6:

[Figure]

**Figure 6.** Elevation-based delineation of three zones in BJRW (a) and NTRW (b), classified using DEM data as Zone 1 (0-40 m), Zone 2 (40-120 m), and Zone 3 (>120 m). Statistical distribution of slopes within these zones illustrated through box plots (c), with mean values labeled numerically.

2. **Based on the SHUD model, the authors analyzed the impact of slope on hydrological processes (subsurface). Local subsurface flow is usually affected by soil characteristics, especially hydraulic conductivity, and is also influenced by slope runoff (slope + accumulating area). Therefore, the groundwater level is higher in the flat areas around the river channel, leading to a larger subsurface flow.**

**REPLY:** We agree with your assessment that local subsurface flow is influenced by soil characteristics and runoff patterns associated with slope and accumulating area (i.e., the topographic wetness index, TWI). As a result, groundwater levels tend to be higher in flatter areas, leading to greater subsurface flow. Our findings support this conclusion. Both the daily and annual subsurface flow results (Fig. 8 and Fig. 9) show a positive correlation between subsurface flow and slope in the flatter Zone 1, indicating that as the slope increases, subsurface outflow also increases in low-elevation areas. However, in the higher-elevation Zones 2 and 3, the relationship differs. Here, the slope is negatively correlated with subsurface outflow. These contrasting patterns highlight the spatial complexity of groundwater flow processes across different topographic conditions. We discuss the implications of these findings in the revised manuscript.

Lines 291-300 (Section 4.2.1):

"In both watersheds, the relationship between slope and subsurface flow varies with elevation, revealing a complex interplay between topography and groundwater dynamics. A negative correlation exists between slope and subsurface flow in Zones 2 and 3, while a positive correlation is observed in Zone 1. This indicates that in the low-elevation Zone 1, as slope increases, subsurface outflow also increases, while in the mid- and high-elevation Zones 2 and 3, as slope increases, subsurface flow decreases. In low-elevation areas, the groundwater table is typically shallow and the soil is relatively saturated. Under these conditions, increasing slope significantly

enhances the lateral hydraulic gradient, thereby facilitating downslope groundwater flow. In mid- to high-elevation areas, the groundwater table is generally deeper. Steeper slopes tend to boost surface runoff, reducing infiltration and diminishing groundwater recharge. Consequently, a negative correlation arises between slope and groundwater outflow in these higher elevation zones."

3.  **The SHUD model outputs subsurface flow, which may include the flow (Q1, Q2, Q3) in three directions of the triangle (positive outward) and the sum/net flow in the three directions (Qs = Q1 + Q2 + Q3, or eleqsub = eleqsub1 + eleqsub2 + eleqsub3). Then, the water balance of the saturated zone of this unit should be dS = Q1 + Q2 + Q3. In the long-term trend, dS should be close to zero. Here, dS is equivalent to Qs (eleqsub). Therefore, subsurface flow should be the sum of positive flows or the sum of negative flows. If the authors directly use new-flow (Qs/eleqsub) in the calculation, it no longer conforms to the meaning of subsurface flow. For example, the flow rates of the three sides of a triangle are eleqsub1 = 200, eleqsub2 = 100, and eleqsub3 = -310, then eleqsub = 200 + 100 + (-310) = -10. It indicates that the unit gains 10 units of water. In this case, the flow rate at this unit should be 300 (200 + 100) or -310, but definitely not -10. Similarly, the calculation of surface runoff may have the same concerns. I suggest that the authors clarify the reading and processing of variables in the appendix or supplementary materials.**

**REPLY:** Firstly, we would like to thank the reviewer for the very clear and detailed explanation. We acknowledge the importance of accurately representing subsurface flow calculations to ensure our results are clear and understandable for readers.

We recognize the concern regarding the use of the net flow value (e.g., -10) instead of positive or negative direction flows (e.g., 300 or -310). In this research, we use the net flow (eleqsub) to represent the water flow amounts within the control volume. While this approach simplifies the explanation of subsurface flow dynamics, it aligns with our study's objective to examine broader hydrological trends and patterns rather than the detailed hydraulic dynamics within each mesh. As mentioned in Appendix B, line 539 of the revised manuscript, we only calculated the triangular meshes with a positive net flow, indicating subsurface or surface net outflow. This ensures our analysis focuses on areas contributing to water movement out of the system, consistent with our objective to better understand outflow dynamics. To clarify the processing of variables, we have included a detailed explanation of the hydrological flow calculation method in the Appendix B.

Lines 541-544 (Appendix B):

"Model outputs for surface runoff and subsurface flow (m³ d⁻¹) represent net flow amounts per mesh grid. For daily analysis, these outputs were summed to total flow volumes (m³) and divided by grid area to obtain flow depths (m). Infiltration and ET outputs (m d⁻¹) were similarly summed to daily depths (m). These standardized depths were used to analyze impacts of slope and LULCC."

4.  **Parameter calibration is a very challenging task for any ISSHMs. Although your study uses manual parameter calibration, I strongly recommend that you share more calibration details/experiences so that others can learn from your work. For example, how were sensitive parameters determined? How were parameters adjusted to gradually approach the observed results? How was it determined that the current parameter set reached "optimal" or "usable" levels?**

**REPLY:** We appreciate the suggestion to provide more details about the parameter calibration process. We have enhanced the description of our parameter calibration process in the revised manuscript. We explain our methodology for identifying the most sensitive parameters, which combined our previous ISSHM calibration

experience, literature insights (Baroni et al., 2010; Song et al., 2015; Liu et al., 2020), and preliminary sensitivity analyses (Section 3.3). This systematic approach led to the selection of seven critical parameters governing both unsaturated zone and aquifer properties (Table 3).

The manual calibration process is an iterative adjustment process, which followed three stages: (1) coarse adjustments to align streamflow trends, peaks, and timing; (2) targeted refinement of aquifer parameters to improve baseflow matching; and (3) fine-tuning of soil and aquifer parameters near monitoring sites to enhance groundwater table simulations while preserving previously established parameter values. These stages were iteratively repeated until meeting performance criteria (NSE > 0.5 and groundwater tables within acceptable observational ranges), ensuring the robustness and reliability of the final parameter set. In the revised manuscript, we have included a comprehensive description of our manual parameter calibration approach.

Lines 177-192 (Section 3.3):

"Given the heterogeneity of the basin and the calibration target covering two types and multiple sites of monitoring datasets, effective automatic calibration becomes extremely difficult. Therefore, manual calibration methods are often preferred for ISSHMs (Shi et al., 2014; Thornton et al., 2022; Brandhorst and Neuweiler, 2023). Monitoring data from the entire period were utilized for calibration, focusing on enhancing model performance. Parameter selection was guided by prior ISSHM calibration experience, insights from the literature (Baroni et al., 2010; Song et al., 2015; Liu et al., 2020), and preliminary sensitivity analyses. Informed by these combined efforts, we identified seven critical parameters related to unsaturated zone and aquifer properties for calibration (Table 3).

As the calibrated parameters were not independent, an iterative adjustment process was required. Initially, all parameters were coarsely adjusted to match the simulation river flow with monitoring data, emphasizing trends, peak timing, and peak values, even though consistency in baseflow simulation results was not yet achieved. The next stage focused primarily on modifying aquifer-related parameters to ensure that the simulated baseflow closely matched the monitoring results. In the final stage, the groundwater table was calibrated by refining soil and aquifer parameters near the monitoring sites while minimizing significant changes to previously established parameters. These three stages were repeated until the model met our performance criteria, defined as achieving a Nash-Sutcliffe Efficiency (NSE) for streamflow greater than 0.5 and simulated groundwater tables falling within acceptable observational ranges. A detailed discussion of the final calibrated parameters and results is provided in Sect. 4.1."

5. **Although the authors have clarified the data sources of this study, I strongly recommend that the authors share the research data of this article (including model input files, model source code used to implement this study, and calibrated parameters) in an open database (such as Zenodo). Of course, you can desensitize or not share meteorological and hydrological data that are restricted by copyright or confidentiality clauses. Sharing this data can help other researchers replicate your work and further expand the impact of your work. This is just a suggestion.**

**REPLY:** Thank you for your thoughtful suggestion regarding data sharing. We have shared the model input files, element and river shape files, along with land cover datasets in Zenodo (DOI: 10.5281/zenodo.14539888).

**References in the REPLY part**

Baroni, G., Facchi, A., Gandolfi, C., Ortuani, B., Horeschi, D., and van Dam, J. C.: Uncertainty in the determination of soil hydraulic parameters and its influence on the performance of two hydrological models of different

complexity, Hydrol. Earth Syst. Sci., 14, 251–270, https://doi.org/10.5194/hess-14-251-2010, 2010.

Liu, H., Dai, H., Niu, J., Hu, B. X., Gui, D., Qiu, H., Ye, M., Chen, X., Wu, C., Zhang, J., and Riley, W.: Hierarchical sensitivity analysis for a large-scale process-based hydrological model applied to an Amazonian watershed, Hydrol. Earth Syst. Sci.,24, 4971–4996, https://doi.org/10.5194/hess-24-4971-2020, 2020.

Song, X., Zhang, J., Zhan, C., Xuan, Y., Ye, M., and Xu, C.: Global sensitivity analysis in hydrological modeling: Review of concepts, methods, theoretical framework, and applications, J. Hydrol., 523, 739-757, https://doi.org/10.1016/j.jhydrol.2015.02.013, 2015.

**Referee #2**

**General comments:**

This article is a comparative analysis of two adjacent watersheds using the integrated surface-subsurface hydrological model (ISSHM) to examine the effects of topography and land use/land cover change (LULCC) on hydrological processes within the Greater Bay Area (GBA) of China. In general, the datasets used and methodology for the analyses are clear and appropriate. The results are well-organized. In the conclusion part, by mentioning the importance for hydrologic management strategies to consider the specific topography and LULCC characteristics of each watershed, the high-level significance of the study stands out.

However, if the significance of the study could be fully investigated with more in-depth thoughts about the results and conclusion, the publication would represent a more substantial contribution to scientific progress in the field.

The four scenarios using HLU and CLU seems intuitive and simple to apply, which might become a lightly novel or highlight of the methodology that could be potentially widely-applied. However, insufficient details about "all built-type land uses" and the process of "The HLU pattern involves reverting all built-type land uses in both watersheds to their pre-construction conditions" are provided. This is not addressed explicitly in the discussion or supplementary, which could become a valuable contribution to future studies.

Moderate to major revision is suggested to this manuscript.

**REPLY:** Thank you for your comprehensive and detailed evaluation of our manuscript. Your valuable comments on the importance of this research are of great significance to us. In response to your feedback, we have made some significant revisions to improve both the depth of our analysis and the overall structure of the manuscript. Specifically, in the Methods section, we have re-described the land use patterns and created a separate subsection (Section 3.4.1) to provide a clearer and more detailed explanation. In the Results section, for instance, we have conducted statistical analyses on areas where significant hydrological behavior changes occurred caused by urbanization and summarized these findings to better illustrate their impact on hydrological processes (lines 348-354, Figure 10). Additionally, we have expanded the Discussion section to offer a more thorough exploration of our results and their implications for hydrologic management.

Detailed responses to your specific comments are provided in the subsequent section. We hope these changes effectively address your concerns and enhance the manuscript.

Lines 348-354 (Section 4.2.3):

"Regions with a KS statistic greater than 0.5 are considered to be significantly affected by urbanization. The spatial statistical characteristics of these regions for four hydrological processes are analyzed in Figs. 10b–d. Infiltration exhibits the most extensive spatial impact, whereas changes in surface runoff, subsurface flow, and ET are confined to more limited areas (Fig. 10b). Considering the elevation variations, the influenced surface runoff and ET regions are more significant at higher elevations, while the most influenced subsurface flows are limited to lower elevation regions (Fig. 10c). Notably, areas with significant ET changes are characterized by steeper slopes (Fig. 10d). Figure 10 demonstrates that hydrological processes most influenced by urbanization are not uniform but rather concentrated in specific regions."

Figure 10:

[Figure]

**Figure 10.** Spatial analysis of urbanization impacts on hydrological processes in NTRW and BJRW. (a) Spatial distribution of KS statistics and absolute differences between first and second scenarios for four hydrological processes: surface runoff (Surf), subsurface flow (Sub), evapotranspiration (ET), and infiltration (Infil). The color scale represents the absolute value of differences, with areas outlined in pink and red indicating KS statistics > 0.5 and > 0.75, respectively. (b) Percentage of significantly affected areas (KS > 0.5) for each hydrological process. (c) Elevation distribution and (d) slope distribution of significantly affected areas, with blue and red boxes representing NTRW and BJRW, respectively. Box plots show the median (horizontal line), 25th and 75th percentiles (box boundaries), and the mean value (white dot with corresponding text above each box).

**Specific comments:**

- **Introduction:**

1. **L73-74: consider adding reference to support the statement "they are mainly based on single and spatially homogeneous watersheds".**

    **REPLY:** Thank you for your valuable suggestion. We have revised this statement and added relative literature.

    Lines 75-77 (Section 1):

"While some studies have investigated the effects of LULCC and topography using the ISSHM approach, they are primarily based on the single watershed (Chu et al., 2010; Im et al., 2009; Thanapakpawin et al., 2007), hindering comparative analyses."

2. **L77: may consider adding one or two sentences to describe "crucial economic zone" in China to highlight the importance of the study area.**

**REPLY:** Thank you for your valuable suggestion. We have added a description of the Greater Bay Area (GBA) to emphasize its significance as a critical economic zone.

Lines 78-83 (Section 1):

"We simulate the hydrological processes of two watersheds in the Greater Bay Area (GBA), a critical economic zone in China that encompasses major cities such as Guangzhou, Shenzhen, Hong Kong, and Macao. According to official data, the GDP of the GBA exceeded 14 trillion yuan in 2023 (Greater Bay Area, 2024). Despite this economic success, the region faces significant challenges in achieving sustainable growth under rapid urbanization, making it an ideal case study for investigating the impacts of development on hydrological processes."

- **Methodology:**
1. **In section 3.4, the four scenarios using HLU and CLU seems intuitive and straightforward to apply, which might become a highlight of the methodology that could be potentially widely-applied. However, insufficient details about "all built-type land uses" and the process of "The HLU pattern involves reverting all built-type land uses in both watersheds to their pre-construction conditions" are provided. This is not addressed explicitly in the discussion or conclusion, and could be a valuable contribution in future studies.**

**REPLY:** Thank you for your valuable comments. Given that the study focuses on land use change and terrain slope, we have restructured Section 3.4 in the revised manuscript by dividing it into two subsections. In Section 3.4.1, we provide more detailed explanations of the two land use patterns.

Lines 206-223 (Section 3.4.1):

"**3.4.1 Two land use patterns**
Among the four scenarios, we implemented two types of land use patterns: Current Land Use (CLU) and Historical Land Use (HLU). The CLU pattern was derived from 2020 land use data, which was obtained from the Dynamic World project, with a spatial resolution of 10 meters (Fig. 1d). The CLU pattern was generated by determining the dominant land use type based on areal coverage for each triangular mesh grid and assigning that classification to the corresponding grid (Fig. 5a). To generate the HLU pattern, we modified the CLU pattern by reclassifying all mesh grids identified as built-up land to tree cover in both watersheds, simulating pre-urbanization conditions (Fig. 5b).

Both the original raster data and our hydrological model incorporate eight land use classifications: bare land, crops, shrubs and scrubs, grassland, flooded vegetation, trees, built-up land, and water bodies (Fig. 1d and Fig. 5). Each land use type is parameterized with specific values in the model, including leaf area index (LAI), albedo, surface roughness, root zone depth, and impervious surface fraction. The impervious surface fraction is set to 94% for built-up land, as these areas represent high-density urban development. All other land use types are assigned an impervious surface fraction of 0%. Under the CLU pattern, built-up land comprises 37.6% of the NTRW and 69.8% of the BJRW. Following reclassification in the HLU pattern, the built-up land fraction becomes 0% in both watersheds.

[Figure]

**Figure 5.** Model setup of land use patterns for two watersheds: (a) Current Land Use (CLU) pattern showing the present urbanized state with extensive built-up areas (pink) mixed with other land cover types, and (b) Historical Land Use (HLU) pattern representing pre-urbanization conditions, where all built-up areas have been converted back to trees (dark green) to simulate the historical natural state."

2. **L187: consider adding reference of 40 meter, at which Zone 1 and Zone 2 are divided.**

**REPLY:** Thank you for your valuable suggestion. The division of each zone was determined based on two main criteria. First, we calculated the average elevation of each triangular mesh grid using new interpolated DEM raster data. Then, we applied the natural breaks method to classify all grids in the two watersheds into six elevation groups, with the natural breakpoints for the first two groups being approximately 40 meters and 120 meters. We also considered the number of mesh grids within each zone, ensuring that each zone contains enough mesh grids for reliable statistical analysis. Since the number of grids in groups three through six was too small for meaningful analysis, we combined them into a single Zone 3. We have clarified this process in the new subsection 3.4.2 to ensure that this division is clearly explained.

Lines 230-237 (Section 3.4.2):

"For a more detailed examination of slope impacts across different spatial areas within the watersheds, we divided the watersheds into three elevation zones. First, we calculated the average elevation of each triangular mesh grid. Using the natural breaks method, we classified all grids into six elevation groups, with the first and second natural breakpoints at approximately 40 m and 120 m. To ensure sufficient grids for reliable statistical analysis, we grouped the remaining four elevation categories into a single elevation zone. Based on these criteria, we defined three elevation zones:
- **Zone 1** consists of low-elevation grids with DEM values below 40 m, primarily flat regions.
- **Zone 2** includes grids with DEM values from 40 m to 120 m, located at mountain foothills.
- **Zone 3** comprises high-elevation grids with DEM values above 120 m."

3. **L192-203: consider having a flowchart or a table instead of a long paragraph to demonstrate the key factors/processes, could also help the readers better understand the results.**

**REPLY:** Thank you for your helpful suggestion. We have delated the original lines 192-203 instead incorporated a flowchart (Fig. 4) in the revised version of the manuscript.

Lines 199-201 (Section 3.4):

"The overall framework of our assessment methods is illustrated in Fig. 4, with detailed descriptions of land use pattern settings and statistical methods provided in Sects 3.4.1 and 3.4.2, respectively."

Figure 4:

[Figure]

**Figure 4.** Framework for assessing the impacts of slope and LULCC on hydrological processes.

- **Results:**
1. **Figure 8: the text in the x and y-axis are too small to read.**

   **REPLY:** Thank you for pointing this out. We have revised this figure to enlarge the text on both the x and y-axes.

   Figure 11 (formerly Figure 8):

[Figure]

**Figure 11.** Box plots delineating the impacts of LULCC on the four annual outputs across all meshes within each watershed. The comparison contrasts the outcomes under the HLU and CLU patterns. The top row displays the results of NTRW, while the second row displays the results of BJRW. KS test values (C) are annotated, all p-values are less than 0.05.

2. **Figure 9: the current figure is busy with many equations/texts embedded with the dots, may considering make it concise by leaving the p-values in the figure but putting the equations in the captions or texts.**

**REPLY:** Thank you for your valuable suggestion. We have revised the figure by keeping the p-values within the figure and moving the equations to the caption to make the figure less busy and more concise.

Figure 12 (formerly Figure 9):

[Figure]

**Figure 12.** Scatter plots of Spearman statistic values of slope and four model outputs under 29 years of different annual rainfall amounts, with statistical significance levels indicated by p-values in plots. Shaded areas indicate 95% confidence intervals. Regression equations for **surface runoff** (a) NTRB: Zone3 (y=0.03x+0.08); BJRB: Zone1 (y=-0.02x-0.23), Zone3 (y=0.01x+0.3). **Subsurface flow** (b) NTRB: Zone3 (y=-0.05x+0.02); BJRB: Zone1 (y=0.07x+0.09), Zone3 (y=-0.01x-0.35). **ET** (c) NTRB: Zone2 (y=-0.05x+0.16); BJRB: Zone2 (y=0.003x+0.31). **Infiltration** (d) NTRB: Zone3 (y=0.000x-0.17); BJRB: Zone1 (y=0.02x+0.13), Zone2 (y=-0.03x-0.09), Zone3 (y=-0.005x-0.41).

3. **Figure 10: similar to Figure 9.**

**REPLY:** Thank you for your suggestion. We have revised this figure by keeping the p-values in the figure and moving the equations to the caption.

Figure 13 (formerly Figure 10):

[Figure]

**Figure 13.** Scatter plots of KS test coefficients between LULCC and four model outputs under 29 years of different annual rainfall amounts, with statistical significance levels indicated by p-values in plots. Shaded areas indicate 95% confidence intervals. Regression equations for **surface runoff** (a) NTRW (y=0.36-0.003x); BJRW (y=0.62-0.06x). **Subsurface flow** (b) NTRW (y=-0.03+0.08x); BJRW (y=0.06+0.09x). **ET** (c) NTRW (y=0.35+0.01x); BJRW (y=0.55-0.01x). **Infiltration** (d) NTRW (y=0.35+0.001x); BJRW (y=0.63-0.03x).

● **Conclusion:**
1. **L373-379: the author may consider adding more specific examples or related refences about how hydrologic management or local watershed agencies could use this study to improve their methodology and strategies. Thus, the application of this publication could not only benefit not only the future studies in academia but also shed light on the practical water management or engineering world.**

    **REPLY:** Thank you for your valuable suggestion. We have highlighted the practical applications of well-calibrated ISSHM models in land-use scenario design and urban planning, enabling agencies to simulate and assess the impacts of various development patterns on hydrological processes across different temporal and spatial scales.

    Lines 497-500 (Section 5):

"This study underscores the importance of incorporating non-structural approaches in urban water management. Well-calibrated ISSHM models have demonstrated their practical value in land-use scenario design, enabling rapid simulation of how different development patterns affect hydrological processes across temporal and spatial scales. The integration of such hydrological modeling with urban planning will help build more resilient cities."

- **Discussion (e.g., the Limitation and future work section in the current manuscript) can be substantially improved.**

**REPLY:** Thank you for your suggestion. We have substantially enhanced Section 4.4 by restructuring it into three focused subsections:

**Section 4.4.1 Patterns of surface and subsurface hydrological behavior**

This section now offers a detailed analysis of response mechanisms and controlling factors in surface and subsurface hydrology. We examine how these processes respond differently to topography, rainfall intensity, and urbanization.

**Section 4.4.2 Suggestions for urban water resource management**

Building on our findings about hydrological processes and urbanization impacts, we present practical management recommendations. The section addresses watershed-specific flood risks, proposes complementary non-structural approaches, and demonstrates how hydrological modeling, particularly ISSHMs, can inform urban planning decisions.

**Section 4.4.3 Limitation and future work**

We acknowledge the limitations of model calibration, data availability, and geographic specificity. Future research will include uncertainty quantification, analysis of extreme rainfall conditions, and a more detailed examination of ET.

4. **L373-379: the author may consider adding more specific examples or related references about how hydrologic management or local watershed agencies could use this study to improve their methodology and strategies. Thus, the application of this publication could not only benefit not only the future studies in academia but also shed light on the practical water management or engineering world.**

**REPLY:** Thank you for your insightful suggestion. Providing more specific examples and references would significantly enhance the usefulness of our research. In the revised manuscript, we have added a section 4.4.2 to illustrate the impact of these research results on practical water resources management.

Lines 443-463 (Section 4.4.2):

"**4.4.2 Suggestions for urban water resource management**

Urban hydrology is a highly complex issue (McGrane, 2016; Qi et al., 2021). This research indicates that urban hydrological processes are influenced not only by local topography but also by the characteristics of the entire watershed. The effects of urbanization are not uniform but rather distinctly localized, with varying intensities across different spatial areas.

Cities located in steep and rainy watersheds like Hong Kong face more severe challenges. Due to its steep mountainous terrain and limited flat regions, Hong Kong has minimal zones suitable for stable water storage (Chen, 2001). Additionally, with its subtropical monsoon climate bringing intense rainfall during typhoon seasons, Hong Kong faces significant urban flooding risks in its low-elevation, high-density building regions (He et al., 2021; Yang et al., 2022). Although flat cities like Shenzhen have the capability of buffering the effects of urbanization through flatter topography, their high level of urbanization still poses significant challenges for flood management under extreme precipitation conditions. Constrained by space limitations, development has

extended into floodplains, wetlands, and reclaimed coastal zones (Chan et al., 2014).

Evidence suggests that depending exclusively on hard-engineering infrastructure for urban flood defense is both costly and impractical (Chan et al., 2022; Cai et al., 2021). The role of non-structural flood control measures should be emphasized, including public participation and training, the development of comprehensive water resource monitoring networks, and hydrological models for more precise flood monitoring and prediction. Technology-driven warning systems have demonstrated their effectiveness in predicting urban flood risks (Yereseme et al., 2021). The experience of sustainable flood risk management in the UK, Netherlands, USA, and Japan provides useful lessons for developed cities worldwide (Chan et al., 2022). The use of hydrological modeling to combine flood risk assessment with urban planning leads to more resilient urban water management systems. In particular, the application of ISSHMs can greatly enhance predictive capabilities before implementing land-use changes. By calibrating models to reflect current watershed conditions, planners can readily simulate various "what-if" scenarios to evaluate how proposed urban development patterns might alter hydrological processes"

5. **It lacks the in-depth analysis of evapotranspiration (ET), which correlates with climatic factors such as solar radiation, temperature, humidity, etc. It is worth noting that ET is different from surface runoff, subsurface flow, and infiltration. The author may consider adding sentences in the limitation/discussion section.**

**REPLY:** Thank you for your valuable feedback. We admit that the ET process is quite different from surface runoff, subsurface flow and infiltration processes. In addition to land factors, ET is more closely related to climatic factors such as solar radiation, temperature and humidity. Our research focuses mainly on terrestrial hydrological processes, so the discussion of ET is relatively limited. In the revised version, we have included additional discussions of ET when addressing the relationship between slope and annual ET in Zone 2. Furthermore, we have acknowledged the limited discussion of ET in this research within the "Limitations and Future Work" section.

Lines 315-321 (Section 4.2.2):

"In mid-elevation regions (Zone 2), the most significant finding is the positive correlation between annual ET and local slope. This relationship suggests that steeper slopes in mid-elevation zones exhibit higher annual ET amounts. Spearman correlation analysis (results not shown) between slope and annual average soil moisture across Zone 2 grids revealed a correlation coefficient of 0.25 (p-value < 0.05), indicating a positive correlation. Areas with steeper slopes have higher soil moisture, potentially contributing to higher ET amounts. Lee and Kim (2022) reported similar findings in the Sulmachun watershed, Korea, where they observed a positive correlation between surface (10 cm) soil moisture and surface slope through April-December monitoring."

Lines 476-480 (Section 4.4.3):

"Finally, the ET process differs from other three processes as it is influenced not only by land cover but also by climatic factors such as solar radiation, temperature, and humidity (Blyth, 1999). Our findings indicate that establishing a clear, general relationship between topography and ET is difficult. However, the analysis of LULCC and ET shows that converting forested areas into built-up land reduces the total ET at the watershed scale (Fig. 11). Since our research primarily focuses on terrestrial hydrological processes, the discussion of ET remains relatively limited."

6. **Groundwater dynamics was mentioned in other sections except the discussion part, considering address it in section 4.4.**

**REPLY:** Thank you for your valuable suggestion. We acknowledge the importance of groundwater dynamics in our study. In response, we have added subsection 4.4.1 to further summarize and compare the behavioral differences between surface hydrological processes (surface runoff and infiltration) and subsurface hydrological processes. We believe this revised version presents the information more clearly.

Lines 428-442 (Section 4.4.1):

"**4.4.1 Patterns of surface and subsurface hydrological behavior**

Surface and subsurface hydrological processes exhibit distinct differences in their temporal responses and controlling factors. Surface runoff and infiltration respond rapidly and intensely to rainfall events, primarily driven by precipitation at daily timescales, making it difficult to identify stable topographic controls. However, when extending to annual timescales, these quick-response processes gradually reveal their sensitivity to slope and elevation patterns. In contrast, subsurface hydrological processes show weaker direct responses to rainfall, instead relying more heavily on topographic features and upstream water contributions to determine flow patterns.

This research further demonstrates that integrated indicators like the TWI exhibit more pronounced predictive significance for soil moisture patterns at longer (annual) timescales (Seibert et al., 2003; Rinderer et al., 2014; Kopecký et al., 2021). At this temporal scale, soil moisture and groundwater distribution reach a relatively stable state, making topographic influences on both surface and subsurface hydrological processes more evident.

Additionally, urbanization-induced expansion of impervious surfaces has significantly altered surface hydrological processes, with impacts varying across regions and topographic conditions. In contrast to surface processes, urbanization's effects on subsurface flow are less pronounced (Fig. 11), with the most significant changes occurring in low-elevation regions (Fig. 10c), consistent with the findings of Siddik et al. (2022)."

7. **L258-262: consider adding text about the statement that "topographic indices more accurately reflect hydrological responses under steady-state conditions" in discussion part.**

**REPLY:** Thank you for your insightful suggestion. We agree that elaborating on how "topographic indices more accurately reflect hydrological responses under steady-state conditions" will strengthen our discussion. In addition to adding related content in the "4.4 Further discussion" section, we also include the relevant discussion immediately following the results.

Lines 310-314 (Section 4.2.2):

"Seibert et al. (2003) and Rinderer et al. (2014) noted that topographic indices more accurately reflect hydrological responses under steady-state conditions. Specifically, Rinderer et al. (2014) reported from their analysis of data from 51 groundwater wells in a Swiss catchment that the ability of the TWI to predict water table distributions diminishes under unsteady conditions. These findings from previous studies align with our results, where the stronger correlations observed at annual (more steady-state) scales compared to daily (unsteady) scales suggest that topographic controls on hydrological processes are more pronounced and predictable over longer time periods when the system approaches steady-state conditions."

Lines 435-438 (Section 4.4.2):

"This research further demonstrates that integrated indicators like the TWI exhibit more pronounced predictive significance for soil moisture patterns at longer (annual) timescales. At this temporal scale, soil moisture and groundwater distribution reach a relatively stable state, making topographic influences on both surface and subsurface hydrological processes more evident."

8. **Considering the study areas are the important economic zone, the author may consider relating it to other economic zone or highly-urbanized areas in other regions of the world. It may worth thinking and adding texts about how this publication could shed light on the practical water management or engineering world.**

**REPLY:** Thank you for suggesting the connection of our findings to other major urban regions globally. We have expanded Section 4.4.2 to include broader implications for urban water management. By referencing comparable hydrological challenges and management approaches in regions such as the United States, Europe, and other Asian megacities, we aim to offer a broader, more globally applicable perspective.

Lines 447-463 (Section 4.4.2):

"Cities located in steep and rainy watersheds like Hong Kong face more severe challenges. Due to its steep mountainous terrain and limited flat regions, Hong Kong has minimal zones suitable for stable water storage (Chen, 2001). Additionally, with its subtropical monsoon climate bringing intense rainfall during typhoon seasons, Hong Kong faces significant urban flooding risks in its low-elevation, high-density building regions (He et al., 2021; Yang et al., 2022). Although flat cities like Shenzhen have the capability of buffering the effects of urbanization through flatter topography, their high level of urbanization still poses significant challenges for flood management under extreme precipitation conditions. Constrained by space limitations, development has extended into floodplains, wetlands, and reclaimed coastal zones (Chan et al., 2014).

Evidence suggests that depending exclusively on hard-engineering infrastructure for urban flood defense is both costly and impractical (Chan et al., 2022; Cai et al., 2021). The role of non-structural flood control measures should be emphasized, including public participation and training, the development of comprehensive water resource monitoring networks, and hydrological models for more precise flood monitoring and prediction. Technology-driven warning systems have demonstrated their effectiveness in predicting urban flood risks (Yereseme et al., 2021). The experience of sustainable flood risk management in the UK, Netherlands, USA, and Japan provides useful lessons for developed cities worldwide (Chan et al., 2022). The use of hydrological modeling to combine flood risk assessment with urban planning leads to more resilient urban water management systems. In particular, the application of ISSHMs can greatly enhance predictive capabilities before implementing land-use changes. By calibrating models to reflect current watershed conditions, planners can readily simulate various "what-if" scenarios to evaluate how proposed urban development patterns might alter hydrological processes."

9. **The abstract is also suggested to revise based on the updated revision.**

**REPLY:** Yes! We have revised the abstract according to the updated manuscript.

Lines 8-9 (Abstract):

"In economically advanced regions, coordinating land use planning and water resource management is essential for mitigating flood risks and ensuring sustainable development."

Lines 21-23 (Abstract):

"Overall, ISSHM provides robust analysis of LULCC effects on watershed hydrology across scales, enabling predictive approaches to optimize urban management for sustainable development in growing cities."